# Structure of the Rpn13-Rpn2 complex provides insights for Rpn13 and Uch37 as anticancer targets

Xiuxiu Lu[1], Urszula Nowicka[1], Vinidhra Sridharan[1], Fen Liu[1], Leah Randles[1], David Hymel[2], Marzena Dyba[3,4], Sergey G. Tarasov[4], Nadya I. Tarasova[5], Xue Zhi Zhao[2], Jun Hamazaki[6], Shigeo Murata[6], Terrence R. Burke, Jr.[2] & Kylie J. Walters[1]

Proteasome–ubiquitin receptor hRpn13/Adrm1 binds and activates deubiquitinating enzyme Uch37/UCHL5 and is targeted by bis-benzylidine piperidone RA190, which restricts cancer growth in mice xenografts. Here, we solve the structure of hRpn13 with a segment of hRpn2 that serves as its proteasome docking site; a proline-rich C-terminal hRpn2 extension stretches across a narrow canyon of the ubiquitin-binding hRpn13 Pru domain blocking an RA190-binding surface. Biophysical analyses in combination with cell-based assays indicate that hRpn13 binds preferentially to hRpn2 and proteasomes over RA190. hRpn13 also exists outside of proteasomes where it may be RA190 sensitive. RA190 does not affect hRpn13 interaction with Uch37, but rather directly binds and inactivates Uch37. hRpn13 deletion from HCT116 cells abrogates RA190-induced accumulation of substrates at proteasomes. We propose that RA190 targets hRpn13 and Uch37 through parallel mechanisms and at proteasomes, RA190-inactivated Uch37 cannot disassemble hRpn13-bound ubiquitin chains.

[1] Protein Processing Section, Structural Biophysics Laboratory, Center for Cancer Research, National Cancer Institute, Frederick, Maryland 21702, USA. [2] Chemical Biology Laboratory, Center for Cancer Research, National Cancer Institute, Frederick, Maryland 21702, USA. [3] Basic Science Program, Leidos Biomedical Research, Inc., Structural Biophysics Laboratory, Frederick National Lab, Frederick, Maryland 21702, USA. [4] Biophysics Resource, Structural Biophysics Laboratory, Center for Cancer Research, National Cancer Institute, Frederick, Maryland 21702, USA. [5] Cancer and Inflammation Program, Center for Cancer Research, National Cancer Institute, Frederick, Maryland 21702, USA. [6] Laboratory of Protein Metabolism, Graduate School of Pharmaceutical Sciences, University of Tokyo, Bunkyo-ku, Tokyo 113-0033, Japan. Correspondence and requests for materials should be addressed to K.J.W. (email: kylie.walters@nih.gov).

The ubiquitin–proteasome system (UPS) performs regulated protein degradation in eukaryotes through a multistep process by which protein substrates are first modified with ubiquitin chains and subsequently delivered to proteasomes for proteolysis[1]. Substrate degradation at proteasomes occurs within a hollow catalytic chamber at the centre of its 20S core particle (CP). Ubiquitinated substrates are recognized by a 19S regulatory particle (RP) that caps the CP to form the proteasome holoenzyme, as reviewed in refs 2,3. The UPS is essential, ensuring orderly cell cycle progression, signal transduction, clearance of damaged proteins and maintenance of general protein homeostasis. Dysfunction in the UPS is associated with various diseases, as reviewed in refs 4,5, with hyperactivation of proteasome function often invoked by cancer cells[4,6,7] and inhibitors specifically targeting the CP (bortezomib, carfilzomib and ixazomib) used clinically to treat haematological cancers[8–10].

Three receptors in the RP (Rpn1/S2/PSMD2, Rpn10/S5a/PSMD4 and Rpn13/Adrm1) capture substrates by binding to ubiquitin and substrate shuttle factors[11–20]. Dss1/Sem1 is also reported to bind ubiquitin[21], but it is unclear whether this protein, which localizes to the proteasome lid[22], functions as a ubiquitin receptor in the proteasome[18]. Following capture, ubiquitin chains are disassembled by deubiquitinating enzymes of the RP, namely Rpn11/PSMD14, Ubp6/Usp14 and Uch37/UCHL5, while substrates are unfolded and translocated into the CP by a hexameric ATPase ring, as reviewed in refs 2,3. The substrate receptors of the proteasome are structurally distinct, with hRpn10 docked into the RP by an N-terminal von Willebrand factor A domain, while two helical ubiquitin interacting motifs orient as needed to bind ubiquitin chains[23,24]. The proteasome/cyclosome (PC) repeat protein hRpn1 has two recognition regions for ubiquitin-fold molecules, one distinct for substrates and the other for Ubp6/Usp14 (ref. 18). hRpn13 binds ubiquitin chains with loops from an N-terminal Pru (pleckstrin-like receptor for ubiquitin) domain[16,17] that also integrates hRpn13 into the proteasome by binding to 106 kDa PC repeat proteasome subunit hRpn2/S1/PSMD1 (refs 25–28). Although cryoEM-based structures of the 26S proteasome have emerged, the region where Rpn13 localizes remains poorly characterized[29,30]. The spherical shape of scRpn13 has prevented assignment of a defined orientation in cryo-electron microscopy (cryoEM) density maps[16]. High-resolution cryoEM structures of human 26S proteasome have recently been reported by two independent groups[31,32], but with no information for hRpn13. Thus, the mechanism for hRpn13 incorporation into the proteasome remains unclear.

hRpn13 may couple substrate recruitment with deubiquitination through a C-terminal DEUBAD (DEUBiquitinase ADaptor) domain that binds Uch37 (refs 27,28,33). hRpn13 activates Uch37 catalytic activity[28,33] and recent crystal structures of the Rpn13[DEUBAD]-Uch37 complex suggest that the DEUBAD domain promotes accessibility of the Uch37 active site to ubiquitin[34–36]. The two functional domains in hRpn13 interact in an intramolecular fashion, reducing affinity for ubiquitin[37]. Interaction with hRpn2 and the proteasome activates hRpn13 for ubiquitin binding by releasing the autoinhibitory Pru:DEUBAD interaction[37]. The DEUBAD domain of free hRpn13 adopts an 8-helical bundle[37] that splits to engulf a region in Uch37 that is C-terminal to its catalytic domain and unique to this deubiquitinating enzyme[34–36]. In vitro, Rpn10 and Rpn13 can bind to a common K48-linked diubiquitin[24], and may coordinately recruit ubiquitinated proteins to proteasomes.

In recent years, Rpn13 has emerged as a therapeutic target for cancers, including bortezomib-resistant multiple myeloma[38]. The bis-benzylidine piperidone derivative RA190 was found to restrict multiple myeloma and ovarian cancer xenografts[38], and to act synergistically with lenalidomide, pomalidomide or bortezomib against multiple myeloma[39]. Another study independently found that an Rpn13-targeting peptoid inhibitor exerts selective cytotoxicity for multiple myeloma cells[40]. Several findings substantiate a role for hRpn13 in human cancers. hRpn13 mRNA levels are elevated in colorectal[41], ovarian[42] and gastric[43] cancers, and cellular proliferation and migration are inhibited, with apoptosis induced in cell lines derived from these cancers by knock down of hRpn13 (refs 41,44,45). Moreover, hRpn13 and Uch37 are each essential for robust cell cycle progression in HeLa cells[46].

Herein, we define how hRpn13 is assembled into the RP by solving the structure of the hRpn13 Pru domain in a complex with the region of hRpn2 to which it binds in the proteasome. This structure in combination with mechanistic studies provides insights that challenge the current model for the mechanism of action of hRpn13-targeting molecule RA190. Currently approved proteasome inhibitors all target the same enzymatic activity in the proteolytic CP. Our findings highlight an inhibitory mechanism that occurs at a different proteasome location than that currently targeted.

## Results

**Structure of hRpn13 at the proteasome.** The C-terminal 38 amino acids of hRpn2 are sufficient for interaction with hRpn13 (refs 17,47,48). To further define the hRpn13-binding region in hRpn2, we generated smaller fragments and assayed for binding by isothermal titration calorimetry (ITC) to hRpn13 (1–150) which includes the Pru domain. A dissociation constant ($K_d$) of $27 \pm 10$ nM was found for the binding of hRpn2 (940–953) to hRpn13 Pru (Table 1 and Supplementary Fig. 1a). Further truncation to hRpn2 (944–953) impaired binding, with an increased $K_d$ value of $1.96 \pm 0.22$ μM (Table 1 and Supplementary Fig. 1a). A strong interaction between hRpn13 Pru and hRpn2 (940–953) was also indicated by measurements of thermal stability. Label-free differential scanning fluorimetry (DSF) indicated a shift in melting temperature for hRpn13 Pru from $44.6 \pm 0.5$ to $59.4 \pm 0.3$ °C upon binding hRpn2 (940–953) (Fig. 1a). We also used fluorescence polarization (FP) to measure the binding affinity between full-length hRpn13 and hRpn2 (940–953). This approach yielded a $K_d$ value of $14.7 \pm 0.6$ nM (Fig. 1b), indicating that the hRpn13 DEUBAD domain does not impair hRpn13 Pru binding to hRpn2.

To verify that hRpn2 (940–953) can interact with hRpn13 in a cellular context, we expressed FLAG-EGFP-hRpn2 (940-953), FLAG-EGFP-hRpn2 (940-947) or FLAG-EGFP (control) in HCT116 cells. Anti-FLAG antibodies immunoprecipitated endogenous hRpn13 with FLAG-EGFP-hRpn2 (940-953) (Fig. 1c, lane 4), but not FLAG-EGFP (control) (Fig. 1c, lane 2) or FLAG-EGFP-hRpn2 (940-947) (Fig. 1c, lane 3).

Having defined the binding interaction biochemically, we next used nuclear magnetic resonance (NMR) spectroscopy to solve the structure of hRpn13 Pru complexed with hRpn2 (940–953), as described in Methods. In total, chemical shift values were assigned to 94 and 93% of the hRpn13 Pru (spanning N20 to N130) and hRpn2 (940–953) atoms respectively in this complex. Our hRpn13 construct spanned amino acids M1-L150, but the

**Table 1 | Dissociation constants for hRpn13 Pru with hRpn2-derived peptides.**

| hRpn2 | $K_d$ (μM) |
|---|---|
| 940–953 | $0.027 \pm 0.010$ |
| 944–953 | $1.96 \pm 0.22$ |

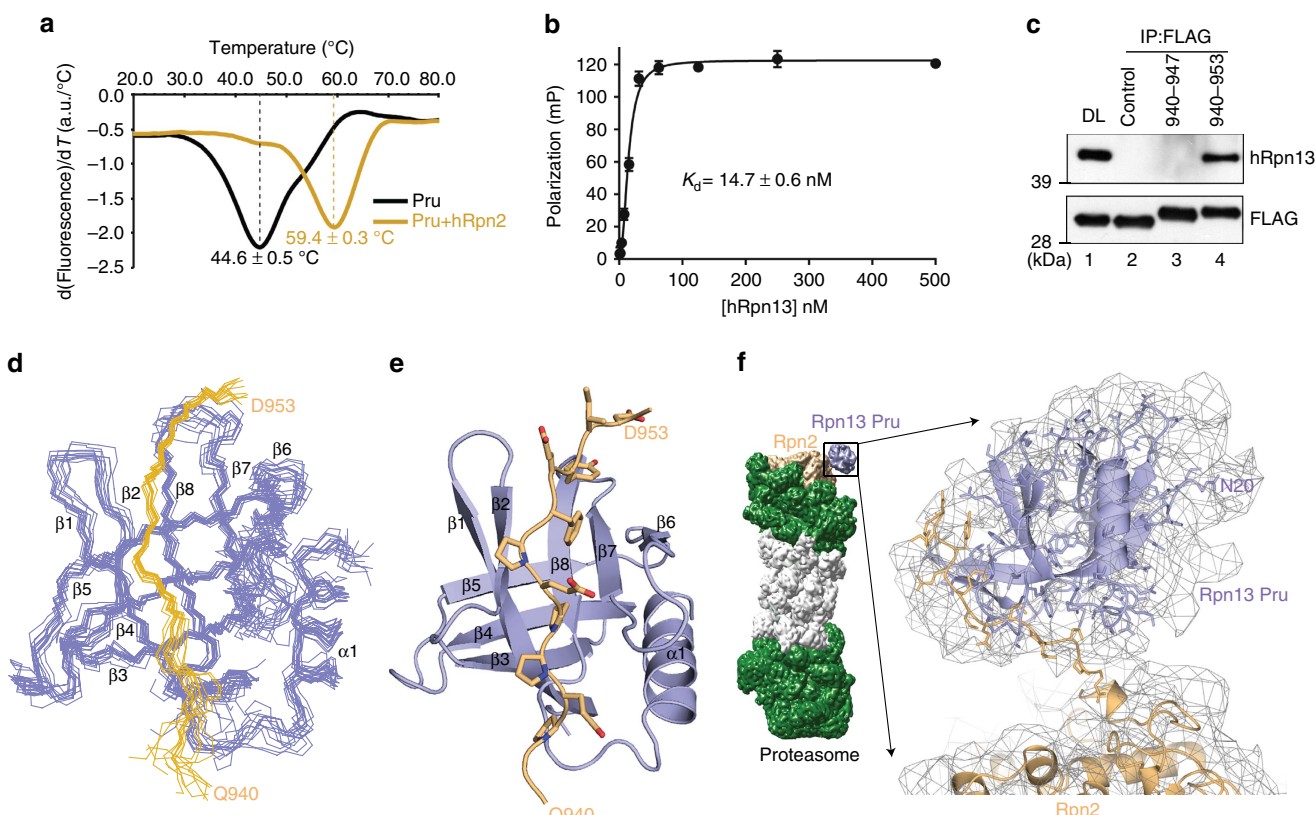

**Figure 1 | Structure of hRpn13 at the proteasome.** (**a**) hRpn13 Pru alone (black) or with hRpn2 (940–953) (orange) was heated from 20 to 85 °C at a rate of 1 °C per min. The melting temperature was calculated from the first derivative of tryptophan emission intensities at 350 nm (a.u., arbitrary unit). (**b**) 10 nM FITC-labelled hRpn2 (940–953) was incubated with varying concentrations of full-length hRpn13 as indicated in triplicate to measure a binding constant by FP. Corresponding values for the probe alone were subtracted from measurements of the complex and the final numbers plotted against hRpn13 concentration and fit by nonlinear regression to a Hill slope model. The error bar represents the s.d. of each data point to the average value. (**c**) Cell lysates or immunoprecipitates derived by anti-FLAG antibodies from HCT116 cells expressing FLAG-EGFP (control), FLAG-EGFP-hRpn2 (940–947) or FLAG-EGFP-hRpn2 (940–953) were subjected to immunoprobing, as indicated. Direct loading (DL) of the lysates from FLAG-EGFP-expressing HCT116 cells is also included. (**d**) Backbone heavy atoms for the 12 lowest energy structures with best geometry for the hRpn13 Pru-hRpn2 (940–953) complex with hRpn13 displayed in periwinkle blue and hRpn2 in light orange. (**e**) Ribbon diagram for the hRpn13 Pru-hRpn2 (940–953) structure depicting the classic pleckstrin homology fold of hRpn13 Pru (periwinkle blue) with the hRpn2 peptide (light orange) extended across a β-strand surface. hRpn2 nitrogen and oxygen atoms are displayed in blue and red, respectively. (**f**) The hRpn13 Pru-hRpn2 (940–953) structure is modelled into a cryoEM reconstruction (EMD-2594, displayed in grey) from *S. cerevisiae* 26S proteasome[50] that includes the scRpn2 PC repeat region (PDB code 4CR2). The N-terminal 19 amino acids of hRpn13 Pru are randomly coiled and most likely contribute to the extra density displayed near residue N20.

N-terminal 19 and C-terminal 20 amino acids were randomly coiled (Supplementary Fig. 1b), as found previously for free hRpn13 (refs 37,49). A series of NMR experiments were recorded (Supplementary Fig. 2a,b), including half-filtered nuclear Overhauser enhancement spectroscopy (NOESY) experiments, to define unambiguous intermolecular interactions between hRpn13 Pru and hRpn2 (940–953), as we did previously to solve the structure of the Rpn1-ubiquitin complex[18]. In total, 140 unambiguous intermolecular distance constraints were identified and used to solve the structure (Table 2 and Supplementary Table 1). The 12 lowest energy structures with best geometry converged to a backbone root mean square deviation of 0.81 Å (Fig. 1d and Table 2).

A representative ribbon diagram for the hRpn13 Pru-hRpn2 (940–953) structure highlights the classic pleckstrin homology fold of hRpn13 Pru, formed by an 8-stranded β-sandwich capped by a C-terminal amphipathic α-helix (Fig. 1e), as was observed for murine[17] and human Rpn13 (ref. 49). The hRpn2 peptide contacts 1,190 Å$^2$ of hRpn13 Pru, capping its β-strand structure, across from the location of the α-helix by binding between β2 and a β-sheet composed of β6 to β8 (Fig. 1e).

Interestingly, in the crystal form of free mRpn13 and hRpn13 Pru, the hRpn2-binding region is occupied by another Rpn13 Pru molecule (Supplementary Fig. 3a) that, similar to hRpn2, buries 1,094 Å$^2$. Residues located on β1, β2 and the β6–β7 loop from one Rpn13 Pru molecule interact with F76 from a neighbouring Rpn13 molecule in a manner akin to their interaction with hRpn2 F948 (Supplementary Fig. 3a). Many rearrangements were observed between the free Rpn13 Pru crystal structures and the hRpn2-bound hRpn13 Pru and their backbone root mean square deviation is 2.65 Å (Supplementary Fig. 3a). The most striking difference is the reconfiguration of β1, β2, and β6 to bend towards hRpn2, like a pincer clamping down on it; the juxtaposed Rpn13 molecule in the crystal requires slightly larger space in this region (Supplementary Fig. 3a and Supplementary Movie 1).

We next sought to use our hRpn13-hRpn2 structure to better define the location of Rpn13 in a cryoEM-based structure of the 26S proteasome. We used sequence alignment to register our hRpn2 fragment to that of *Saccharomyces cerevisiae* (Supplementary Fig. 3b) and manually docked our hRpn13 Pru-hRpn2 (940–953) structure into the cryoEM reconstruction (EMD-2594) with the *S. cerevisiae* Rpn2 structure incorporated

**Table 2 | Structural statistics for the hRpn13 Pru-hRpn2 complex.**

| | hRpn13 Pru (20–130) | hRpn2 (940–953) |
|---|---|---|
| *NMR distance and dihedral constraints* | | |
| Distance restraints | | |
| Total NOE | 1,641 | 141 |
| Intra-residue | 550 | 75 |
| Inter-residue | 1,091 | 66 |
| Sequential ($|i-j| = 1$) | 438 | 47 |
| Nonsequential ($|i-j| > 1$) | 653 | 19 |
| Hydrogen bonds | 45 | 0 |
| Intermolecular NOEs | 140 | |
| Total dihedral angle restraints | | |
| phi | 161 | 18 |
| psi | 161 | 18 |
| | | |
| *Structure statistics* | | |
| Violations (mean and s.d.) | | |
| Distance constraints (Å) | 0.041 | |
| Dihedral angle constraints (°) | 0.390 | |
| Max. dihedral angle violation (°) | 0 | |
| Max. distance constraint violation (Å) | 0 | |
| Deviations from idealized geometry | | |
| Bond lengths (Å) | 0.003 ± 0.000 | |
| Bond angles (°) | 0.454 ± 0.024 | |
| Impropers (°) | 0.330 ± 0.016 | |
| Average pairwise root mean square deviation (r.m.s.d.)* (Å) | | |
| Heavy | 1.55 ± 0.24 | |
| Backbone | 0.81 ± 0.17 | |

*Statistics for secondary structural elements of 12 lowest energy with best geometry structures within hRpn13 Pru (V24-K34, T37-P40, G45-Q51, I57-D63, N68-I75, E81-K83, Y94-K97, R104-W108 and D117-L128) and hRpn2 P942-I951.

(PDB 4CR2)[50] by using UCSF (University of California, San Francisco) Chimera[51]. The resolution for the Rpn13 region of the reconstruction is poor; nonetheless, by fusing the hRpn13-binding region of hRpn2 to the appropriate site in scRpn2, a favoured orientation is suggested for Rpn13 in the density map (Fig. 1f). It is worth noting that the hinge between the Rpn2 region that binds hRpn13 and the preceding toroidal PC repeat domain is undoubtedly flexible. This flexibility would provide conformational freedom for hRpn2-bound hRpn13 Pru domain, facilitating capture of substrates.

**hRpn13 and hRpn2 form extensive and proline-rich contacts.** hRpn2 (940–953) includes four prolines (Supplementary Fig. 3b), all of which interact with hRpn13 amino acids from a trans configuration (Fig. 2a). Strictly conserved P942, P944 and P945 bury hRpn13 W108 (Fig. 2a), as indicated by nuclear Overhauser effect (NOE) interactions (Fig. 2b, upper panel). hRpn2 P942 also interacts with an hRpn13 proline placed at the edge of the interaction surface (P112) and the backbone of Q110 (Fig. 2a). The many interactions involving P942 provide an explanation for the measured reduction in hRpn2 affinity towards hRpn13 upon deletion of Q940 through E943 (Table 1 and Supplementary Fig. 1a). hRpn2 P947 also forms many contacts with hRpn13, interacting with M31, T37, T39 and P40 (Fig. 2a).

In previous work, we found that amino acid substitution of hRpn2 F948 or Y950/I951 results in loss of interaction with hRpn13 (ref. 47). This finding is consistent with the structure of the hRpn13-hRpn2 complex, as hRpn2 F948 and Y950 are buried by many hRpn13 contacts (Fig. 2c). Two hRpn13 valines (V38 and V85) bridge these two hRpn2 aromatic amino acids, while hydrophobic pockets are formed around hRpn2 Y950 by

hRpn13 L33, T36, V95 and R104 and hRpn2 F948 by hRpn13 M31 and V93 (Fig. 2c). These locations in the structure are well defined by NOE interactions from hRpn13 methyl groups to hRpn2 F948 and Y950 (Fig. 2b, lower panel).

**hRpn2 sterically restricts hRpn13 Pru from binding to RA190.** Previous reports indicate that hRpn13 C88 is targeted by RA190 and required for RA190 sensitivity in HCT116 cells[38,39]. Unexpectedly, our hRpn13-hRpn2 structure suggests that hRpn2 sterically blocks the RA190 binding site at C88, as indicated by direct comparison of a model structure of RA190-conjugated hRpn13 Pru (Fig. 3a, left panel) to hRpn2-bound hRpn13 Pru (Fig. 3a, right panel). To test directly whether RA190 reacts with hRpn2-bound hRpn13 Pru, we incubated 20 μM RA190 with 2 μM hRpn13 (1–150) with and without 2 μM hRpn2 (940–953) for 2 h at 4 °C and used mass spectrometry to probe for RA190-conjugated hRpn13 Pru, as described in Methods. hRpn13 contains five cysteines, four in the Pru domain and one in the DEUBAD domain (Fig. 3b). Without hRpn2, the reaction mixture contained species at the correct molecular weight for free and RA190-conjugated hRpn13 Pru (Fig. 3c, black, Table 3 and Supplementary Table 2). However, RA190-conjugated hRpn13 Pru was not detected when this experiment was done with hRpn2 present (Fig. 3c, orange, Table 3 and Supplementary Table 2). This finding is consistent with the hRpn13 Pru-hRpn2 structure and further suggests that RA190 cannot compete with hRpn2 (940–953) for hRpn13 Pru interaction.

We previously demonstrated that RA190 adducts to hRpn13 at the proteasome[38], where the Pru domain is apparently inaccessible (Fig. 3a), but were unable to detect RA190-conjugated DEUBAD domain by NMR in samples that were buffer exchanged by dialysis to remove excess RA190[38]. To resolve this inconsistency, we used liquid chromatography–mass spectrometry (LC-MS) to test whether the covalent bond between RA190 and hRpn13 Pru is labile in the presence of hRpn2 (940–953). We incubated 2 μM hRpn13 Pru with 20 μM RA190 for 1 h at 4 °C and acquired an LC-MS spectrum to find unmodified and RA190-adducted hRpn13 Pru domain (Supplementary Fig. 4, left spectrum). We then in parallel added either 10-fold molar excess hRpn2 (940–953) or an equivalent volume of buffer to yield final concentrations of 2 μM hRpn13 Pru, 20 μM RA190, with or without 20 μM hRpn2 (940–953). LC-MS spectra were recorded on these two mixtures after 1 or 19 h of incubation at 4 °C. A time-dependent reduction in RA190-conjugated hRpn13 Pru was observed by hRpn2 addition, whereas longer incubation times allowed this species to increase when hRpn2 was not present (Supplementary Fig. 4). This result indicates that RA190 reacts reversibly with hRpn13 Pru and is displaced by hRpn2. Such reversibility is also reported for b-AP15 (refs 52,53) that is chemically similar to RA190.

We hypothesized that RA190 could be even more labile towards the hRpn13 DEUBAD domain, as the Pru domain provides a binding pocket for RA190 when it is conjugated to C88 (ref. 38). By using optimized conditions, including more stringent removal of reducing agent, more diluted samples and retaining RA190 in the reaction mixture, we detected RA190 conjugated to hRpn13 DEUBAD (Fig. 3d, Table 3 and Supplementary Table 2). Moreover, we found that full-length hRpn13 interacts with RA190 when hRpn2 (940–953) is present. Without hRpn2, up to three RA190 molecules can conjugate to hRpn13 (Fig. 3e, black, Table 3 and Supplementary Table 2). In contrast, only one RA190 molecule conjugates to hRpn2-bound hRpn13 (Fig. 3e, orange, Table 3 and Supplementary Table 2). Altogether, these findings indicate that the DEUBAD domain, and not the Pru domain, is accessible to RA190 in the presence of hRpn2.

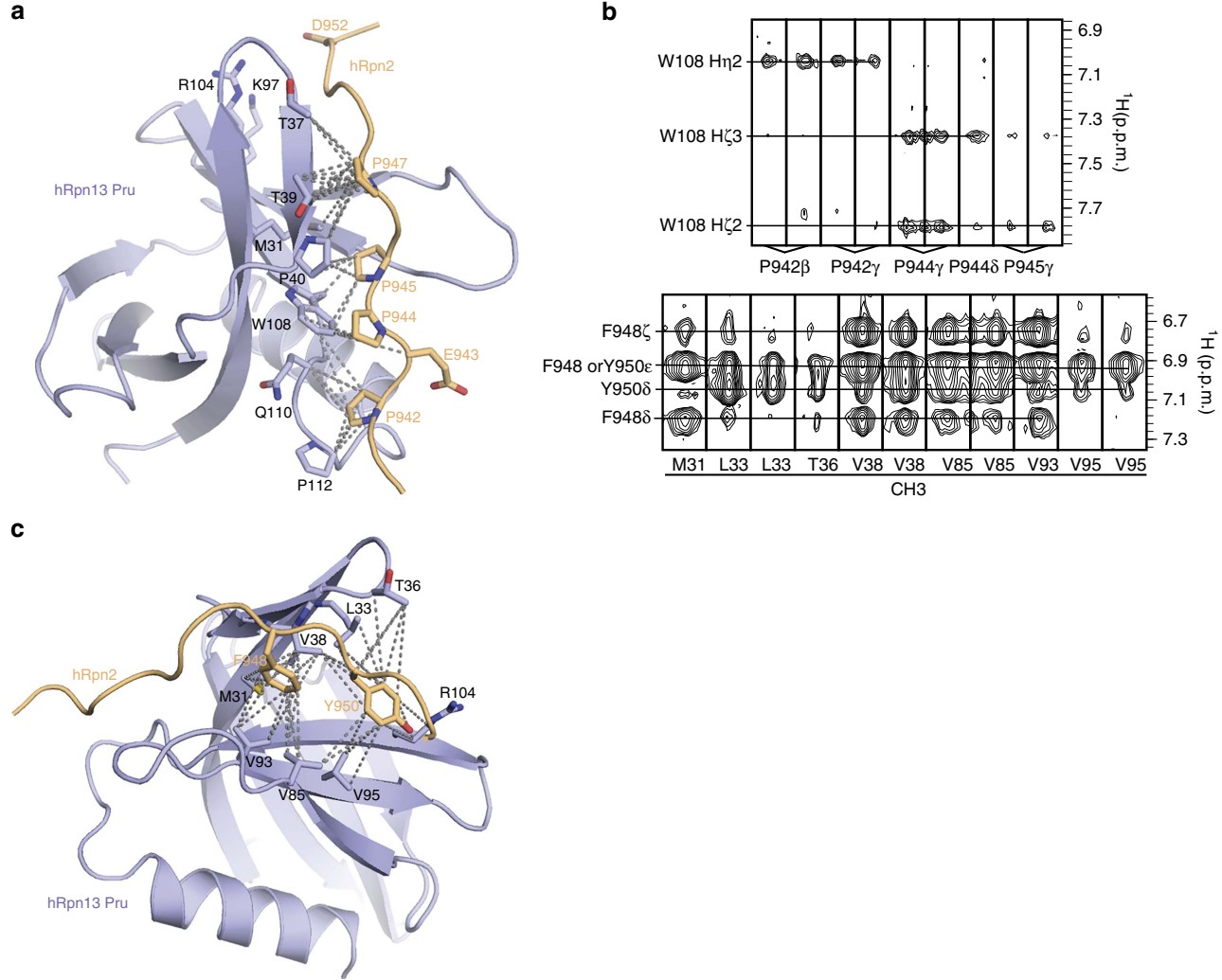

**Figure 2 | hRpn2 zippers along an hRpn13 surface with extensive interactions and a proline-rich contact surface. (a,c)** Ribbon diagram of hRpn13 (periwinkle blue) with heavy atoms at the hRpn2 contact surface displayed. All hRpn2 heavy atoms are shown (light orange) with dashed lines representing intermolecular NOE interactions involving (**a**) hRpn2 P942-P945 and P947 and (**c**) hRpn2 F948 and Y950. Nitrogen, oxygen and sulfur side-chain atoms are displayed in blue, red and yellow, respectively. The orientation in (**a**) is selected to highlight interactions involving hRpn13 W108. (**b**) Selected intermolecular NOEs between hRpn2 and hRpn13. Selected regions from a $^1$H, $^{13}$C edited NOESY experiment acquired with 0.7 mM $^{15}$N, $^{13}$C-labelled hRpn2 (940–953) and equimolar unlabelled hRpn13 Pru (upper panel) and selected regions from a $^1$H, $^{13}$C half-filtered NOESY experiment acquired with 0.7 mM $^{15}$N, $^{13}$C-labelled hRpn13 Pru and equimolar unlabelled hRpn2 (940–953) (lower panel).

Since substitution of hRpn13 C88 for alanine is reported to cause loss of RA190 sensitivity[39], we used FP to test whether hRpn13 Pru affinity for FITC-hRpn2 (940-953) is reduced following 30 min of incubation with RA190. Inhibition was not observed even in the presence of 8,000-fold molar excess RA190 (Fig. 3f, upper panel). Similarly, the binding affinity between hRpn13 full-length protein and FITC-hRpn2 (940–953) was unaltered by the presence of 100 μM RA190 (Fig. 3f, lower panel).

We next tested whether RA190 affects hRpn13 interaction with proteasomes in HCT116 cells. We immunoprecipitated proteasomes of RA190-treated (1 μM for 24 h) or dimethylsulfoxide (DMSO)-treated cells by using antibodies to hRpt3 (a member of the proteasome ATPase ring) and immunoprobed for the presence of hRpn13. No change in the amount of hRpn13 immunoprecipitated with hRpt3 was observed following RA190 treatment (Fig. 3g, lane 3 versus lane 4). In contrast, performing the same experiment with cells overexpressing FLAG-EGFP-hRpn2 (940–953) demonstrated loss of hRpn13 from proteasomes, an effect not observed by FLAG-EGFP (control) expression (Fig. 3h, lane 4 versus lane 3). Thus, the hRpn2

fragment, but not RA190, effectively competes with endogenous proteasomes for hRpn13. Altogether, our data suggest that binding to proteasomes protects hRpn13 Pru domain from RA190 reactivity.

**RA190 binds Uch37 and inhibits its catalytic activity.** We next tested whether RA190 affects hRpn13 activation of Uch37. RA190 was previously found to have a small inhibitory effect on Uch37 activity when ubiquitin-AMC was used as a substrate, but the effect was considerably reduced compared with established inhibitor Ubal[38]. We therefore performed an *in vitro* assay to directly probe whether RA190 affects deconjugation of K48-linked tetraubiquitin by Uch37. His-Uch37 (1 μM) was incubated for 8 h at 37 °C with 1 μM K48-linked tetraubiquitin (Ub4), 0 or 20-fold molar excess RA190 and no or equimolar hRpn13. Immunoblotting was performed on the reaction mixtures with anti-ubiquitin, anti-His and anti-hRpn13 antibodies. Uch37 activity was evaluated by the production of triubiquitin (Ub3), diubiquitin (Ub2) and monoubiquitin (Ub1).

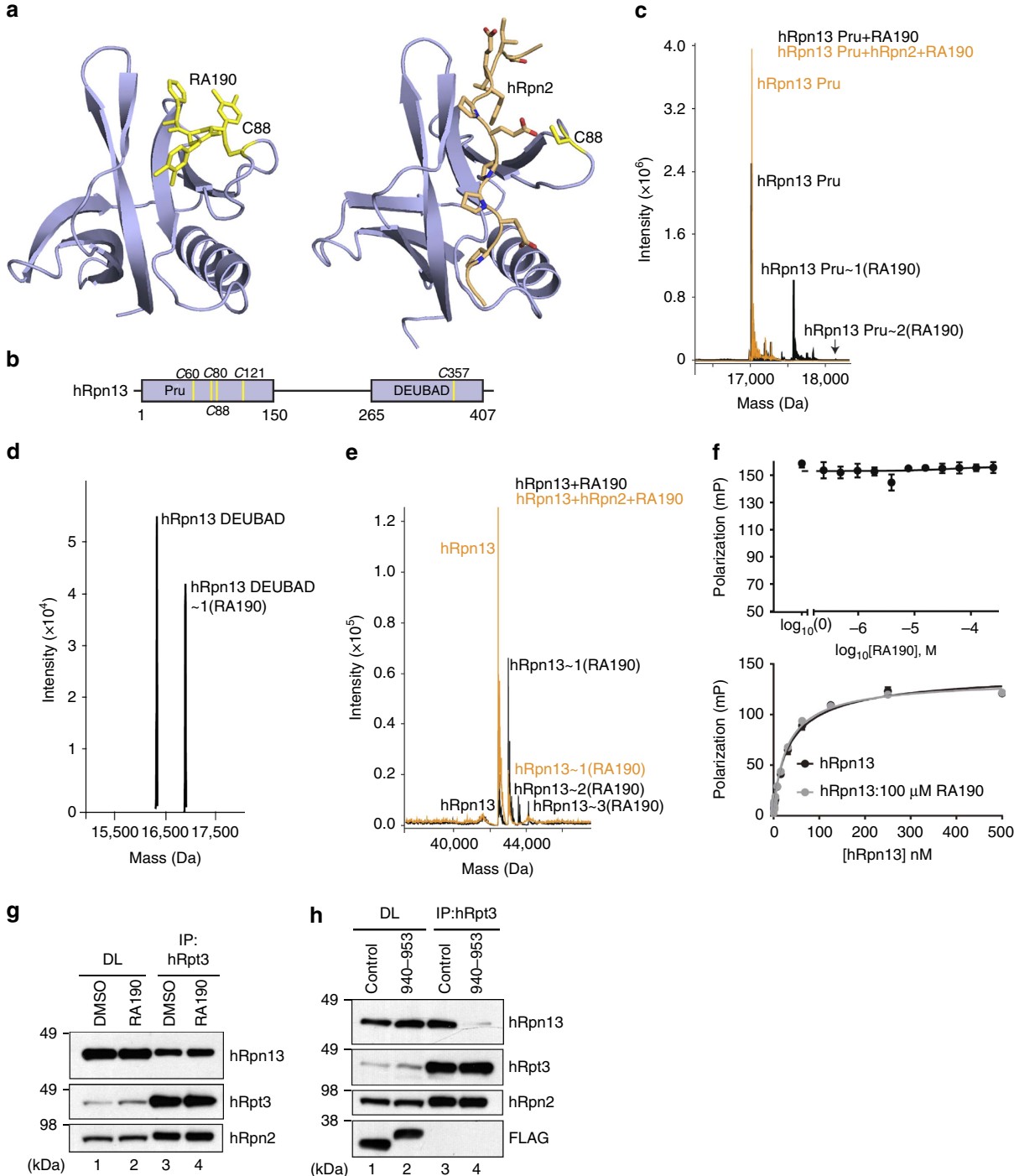

**Figure 3 | hRpn2 restricts RA190 from binding hRpn13 Pru. (a)** Model of RA190 (yellow) adducted to hRpn13 (periwinkle blue) C88 (yellow, left panel) and NMR structure of hRpn13 Pru-hRpn2 (940–953) (right panel). hRpn2 is orange with nitrogen and oxygen in blue and red, respectively. **(b)** Schematic representation of hRpn13 domains with locations of cysteines indicated. **(c–e)** LC-MS analysis of 2 μM hRpn13 **(c)** Pru (black), **(d)** DEUBAD or **(e)** full-length protein (black) or with equimolar hRpn2 (940–953) (orange) following 2 h of incubation with 20 μM RA190. **(f)** FP values for 30 nM hRpn13 Pru incubated serially for 30 min with indicated RA190 concentrations and 10 nM FITC-hRpn2 (940–953) (upper panel). FP values for 100 μM RA190 or 2% DMSO incubated serially for 30 min with varying concentrations of full-length hRpn13 as indicated (lower panel), followed by 10 nM FITC-hRpn2 (940–953). In both cases, FP values were measured in triplicate and corresponding values for FITC-hRpn2 (940–953) alone were subtracted from the measurements. The final values for each panel were plotted against RA190 concentration and fit using the log(inhibitor) versus response–variable, four parameter model (upper panel) or hRpn13 concentration by a Hill slope model (lower panel). The s.d. of each data point to the average value is displayed by error bar. **(g)** HCT116 cells were treated with 1 μM RA190 for 24 h or DMSO (as a control) and the cell lysates immunoprobed (DL) or subjected to immunoprecipitation with anti-hRpt3 antibodies before immunoblotting as indicated. **(h)** Cell lysates (DL) or immunoprecipitates derived by anti-hRpt3 antibodies from HCT116 cells expressing FLAG-EGFP (control) or FLAG-EGFP-hRpn2 (940–953) were subjected to immunoprobing, as indicated.

**Table 3 | Detected hRpn13 species by LC–MS for indicated samples.**

| Sample | hRpn13 species | | | |
|---|---|---|---|---|
| | Free | 1 (RA190) | 2 (RA190) | 3 (RA190) |
| Pru | √ | √ | 0.2% | ND |
| Pru + hRpn2 | √ | ND | ND | ND |
| DEUBAD | √ | √ | ND | ND |
| hRpn13 | √ | √ | √ | √ |
| hRpn13 + hRpn2 | √ | √ | ND | ND |

ND, not detected.
Table summarizing species detected by LC–MS after 2 h of incubation with 10-fold molar excess RA190 (20 µM) at 4 °C for 2 µM hRpn13 (hRpn13), hRpn13 Pru (Pru), hRpn13 DEUBAD (DEUBAD), hRpn13 and hRpn2 (940–953) mixture or hRpn13 Pru and hRpn2 (940–953) mixture. Check marks indicate detected species for unmodified protein (free) or hRpn13 with one (1(RA190)), two (2(RA190)) or three (3(RA190)) RA190 molecules ligated.

Addition of hRpn13 to Uch37 increased the presence of Ub3, Ub2 and Ub1 (Fig. 4a, lane 4 versus 2), as expected[28,33]. In contrast, reduced amounts of Ub3, Ub2 and Ub1 were observed with inclusion of RA190, both in the absence (Fig. 4a, lanes 2 and 3) and presence (Fig. 4a, lanes 4 and 5) of hRpn13. These findings indicate that RA190 inhibits Uch37 activity and, moreover, that it has a direct effect that is independent of hRpn13.

Since RA190 directly affected Uch37 activity, we tested whether it reacts with Uch37 by mass spectrometry, similarly as described in Fig. 3c for hRpn13 Pru. Uch37 contains five cysteines, all localized to its catalytic domain (Fig. 4b). We found RA190 to be highly promiscuous towards Uch37 (Fig. 4c, Table 4 and Supplementary Table 3), adducting to all five cysteines. Moreover, RA190 reacts with Uch37 when hRpn13 and hRpn2 (940–953) are present (Table 4, Supplementary Fig. 5 and Supplementary Table 3). Therefore, our data indicate that RA190 is reactive towards Uch37 and can inhibit its catalytic activity.

**Effect of RA190 at the proteasome.** The hRpn13 DEUBAD cysteine (C357) is directed away from the Uch37-binding surface[34,35]. We thus hypothesized that in cells treated with RA190, hRpn13 would remain competent for interaction with Uch37. To test the impact of RA190 on hRpn13 interaction with Uch37, we evaluated whether Uch37 is crosslinked to hRpn13 in RA190-treated cells, by using a denaturing immunoprecipitation experiment. Lysates from HCT116 cells treated for 24 h with 1 µM RA190 or DMSO (as a control) were incubated with dithiobis(succinimidyl) propionate (DSP) for 30 min followed by lysis in radioimmunoprecipitation (RIPA) buffer. hRpn13 was immunoprecipitated by anti-hRpn13 antibodies and interaction with Uch37 probed by anti-Uch37 antibodies. We consistently observed Uch37 to co-immunoprecipitate with hRpn13 in RA190-treated cells, although a 1.5-fold reduction was detected with RA190 treatment (P value = 0.049, n = 3; Fig. 5a). The region in Uch37 that interacts with hRpn13 is aggregation prone[35,54,55] and it is possible that the small reduction of Uch37 immunoprecipitated with hRpn13 in RA190-treated cells is caused by nonspecific interactions with this region when RA190 is adducted to the Uch37 catalytic domain.

To test further whether RA190 disrupts hRpn13 interaction with Uch37, we used an in vitro pulldown experiment. His-tagged Uch37, hRpn13 or a mixture of equimolar quantities of these two proteins were incubated with 20-fold molar excess RA190 or DMSO (as a control), followed by incubation with Talon resin. After washing with buffer that maintained RA190 or DMSO, retention of hRpn13 on the resin with His-Uch37 was evaluated by SDS–polyacrylamide gel electrophoresis (SDS–PAGE) and

Coomassie staining (Fig. 5b). The result indicated that hRpn13 interacted with Uch37 equivalently when RA190 was present (Fig. 5b, lane 3 versus 2).

Since Uch37 interaction with hRpn13 appeared to be unaffected by RA190, we hypothesized that the presence of Uch37 at the proteasome would similarly be unperturbed by RA190. We tested this hypothesis directly by subjecting cell lysates from DMSO-treated (as a control) or RA190-treated (1 µM, 24 h) HCT116 cells to fractionation over a 10–40% linear glycerol gradient, an established method to isolate proteasomes[56]. This experiment revealed no change following RA190 treatment for the CP component β5, hRpn13 or Uch37 (Fig. 5c); β-actin was included as a cytosolic marker and INO80A, based on its reported interaction with Uch37 (ref. 57). Altogether, our findings led us to conclude that RA190 does not block Uch37 interaction with hRpn13 or the proteasome.

Since Uch37 is expected to disassemble ubiquitin chains at hRpn13 in the proteasome, we hypothesized that RA190 inactivation of Uch37 could impair the disassembly and clearance of ubiquitin chains from the proteasome. To test this model, lysates from HCT116 cells treated as above with RA190 or DMSO were subjected to immunoprecipitation with anti-hRpt3 antibodies followed by immunoprobing with anti-ubiquitin antibodies. RA190 treatment led to increased levels of ubiquitinated protein in the cell lysates (Fig. 5d, lane 2 versus lane 1), as expected[38]. We also found increased levels of ubiquitinated proteins co-immunoprecipitated with hRpt3 following RA190 treatment (Fig. 5d, lane 4 versus lane 3).

We next tested whether RA190 activity at the proteasome requires hRpn13 and Uch37. We used CRISPR-Cas9 gene editing to delete hRpn13 from the HCT116 colon cancer cell line, confirming hRpn13 loss by immunoprobing lysates (Fig. 5e) and performing quantitative PCR (Supplementary Fig. 6) for HCT116 wild-type (WT) and hRpn13-deleted (ΔhRpn13) cells. Proteasomes from lysates of WT and ΔhRpn13 cells treated with RA190 (1 µM for 24 h) or DMSO (as a control) were immunoprecipitated by anti-hRpt3 antibodies and immunoprobed for ubiquitin with anti-ubiquitin antibodies. The presence of ubiquitinated proteins at the proteasome was unaltered by hRpn13 deletion (Fig. 5f, lane 3 versus lane 1), suggesting that proteasome is functional in ΔhRpn13 HCT116 cells. A difference was observed however following RA190 treatment. In particular, no accumulation of ubiquitinated proteins was observed for RA190-treated lysates prepared from ΔhRpn13 cells (Fig. 5f, lane 4 versus lane 3). Consistent with Fig. 5d, ubiquitinated proteins accumulate in RA190-treated HCT116 WT cells (Fig. 5f, lane 2 versus lane 1). Thus, RA190 treatment led to impaired clearance of ubiquitinated proteins at the proteasome through a mechanism that requires hRpn13 and/or Uch37 at the proteasome.

## Discussion

The human Rpn13 orthologue was identified as a proteasome component one decade ago[27,33,58] and soon after, as one of its major ubiquitin receptors[16,17]. Since that time, cryoEM-based structures have emerged of the 26S proteasome but the region where Rpn13 localizes remained poorly characterized[29,30]. We used NMR to define at atomic-level resolution how hRpn13 binds to the proteasome. We find that the extreme C-terminal end of hRpn2 binds an extensive channel formed along an hRpn13 surface in its ubiquitin-binding Pru domain. The hRpn13 Pru-hRpn2 structure challenges the current model for RA190 induction of apoptosis. In particular, the surface that RA190 interacts with when adducted to hRpn13 C88 is occupied by its binding site in the proteasome.

Previous findings demonstrate that RA190 sensitivity is lost upon hRpn13 deletion from HCT116 cells and restored by

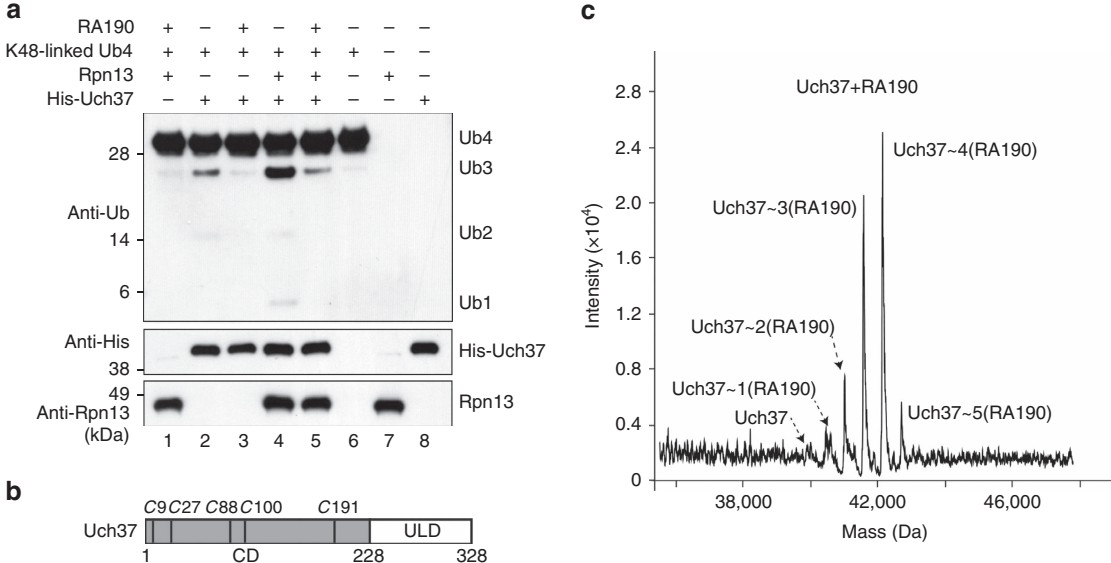

**Figure 4 | RA190 binds Uch37 and inhibits its catalytic activity.** (**a**) *In vitro* deconjugation assay of K48-linked tetraubiquitin (Ub4) by Uch37 with 20-fold molar excess RA190 and/or equimolar hRpn13, as indicated. Immunoprobing was performed as indicated. (**b**) Schematic representation of Uch37 depicting the catalytic domain (CD), hRpn13-binding region (Uch37-like domain (ULD)) and cysteines (italic 'C' with numbers). (**c**) 2 μM purified Uch37 was incubated with 50 μM RA190 for 2 h and samples subjected to LC-MS analysis to detect the formation of RA190 adducts.

**Table 4 | Detected Uch37 and hRpn13 species by LC-MS for indicated samples.**

| Sample | Species | | | | | | | | | |
|---|---|---|---|---|---|---|---|---|---|---|
| | Free | | Uch37∼(RA190) | | | | | hRpn13∼(RA190) | | |
| | hRpn13 | Uch37 | 1 | 2 | 3 | 4 | 5 | 1 | 2 | 3 |
| Uch37 | NA | √ | √ | √ | √ | √ | √ | NA | NA | NA |
| hRpn13 + Uch37 | √ | √ | √ | √ | √ | √ | √ | √ | √ | √ |
| hRpn13 + Uch37 + hRpn2 | √ | √ | √ | √ | √ | √ | √ | √ | ND | ND |

NA, not applicable; ND, not detected.
Table summarizing the species detected by mass spectrometry for the indicated samples; 2 μM Uch37, equimolar mixture of 2 μM hRpn13 and Uch37 or equimolar mixture of 2 μM hRpn13, Uch37, and hRpn2 were incubated with 25-fold molar excess RA190 for 2 h at 4 °C and the samples subjected to LC-MS analysis. Detected Rpn13 or Uch37 species are indicated by a check mark, with the numerical indicator over the column representing numbers of RA190 molecules conjugated based on molecular weight.

expression of wild-type hRpn13, but not by expression of hRpn13 C88A[39]. Although this experiment implicates hRpn13 C88 as being the target of RA190, lack of supporting data including that the C88A mutation does not alter the function of hRpn13 and a lack of direct evidence for RA190 adducting to this cysteine in cells warrants further investigation. If hRpn13 C88 is validated as an essential RA190 target, then this interaction most likely occurs outside of the proteasome. It is possible that newly synthesized hRpn13 may be restricted from assembly into the proteasome by RA190, although we were unable to find evidence for this model *in vitro* (Fig. 3f). It is also possible that hRpn13 performs functions outside of the proteasome that use the RA190-binding site with lower affinity compared with hRpn2. Intriguingly, hRpn13 knockdown leads to reduced levels of Uch37 in cells[27,46] and similar phenotypes are observed by hRpn13 or Uch37 knockdown[27]; however, the mechanism linking hRpn13 to Uch37 protein levels remain unknown.

We report that RA190 reacts with hRpn13 DEUBAD domain C357 (Fig. 3d, Table 3 and Supplementary Table 2); however, this cysteine is directed away from the Uch37-binding surface and does not appear to effect Uch37 interaction with hRpn13 (Fig. 5a,b) or the proteasome (Fig. 5c). Therefore, we do not expect this cysteine to be important for RA190-induced apoptosis, but rather propose that RA190 reaction at this site is a reflection of promiscuity towards exposed cysteine residues.

Thus, RA190 may react with other cellular constituents; however, we find that hRpn13 is required for RA190-induced accumulation of ubiquitinated proteins at the proteasome, suggesting that it does not inhibit hRpn13-independent activities at the proteasome.

Uch37 is contributed to the proteasome by hRpn13 (refs 27,28,33). We propose that RA190 deactivation of Uch37 at the proteasome contributes to induction of apoptosis. Mechanistically, loss of Uch37 activity at the proteasome would cause ubiquitin chains to become stalled at hRpn13 in the proteasome (Fig. 5g, right panel). This model assumes that ubiquitin chain release from hRpn13 is coupled to ubiquitin chain disassembly by Uch37. Our rationale is based on the higher affinity measured for hRpn13 binding to ubiquitin chains compared with monoubiquitin[16]. Consistent with this model, ubiquitinated substrates accumulate at the proteasome in RA190-treated HCT116 cells (Fig. 5d,f) and sensitivity to RA190 is lost following hRpn13 deletion (Fig. 5f). Without hRpn13, ubiquitinated substrates that bind proteasomes by Rpn1 (ref. 18) and Rpn10 (ref. 15) do not rely on Uch37 activity. These two receptors have other nearby deubiquitinating enzymes, namely Usp14 near Rpn1 and Rpn11 near Rpn10, as reviewed in ref. 3; these enzymes are apparently not affected by RA190 in the proteasome context (Fig. 5f). Nonetheless, the presence of ubiquitinated proteins stalled at RA190-inhibited hRpn13-

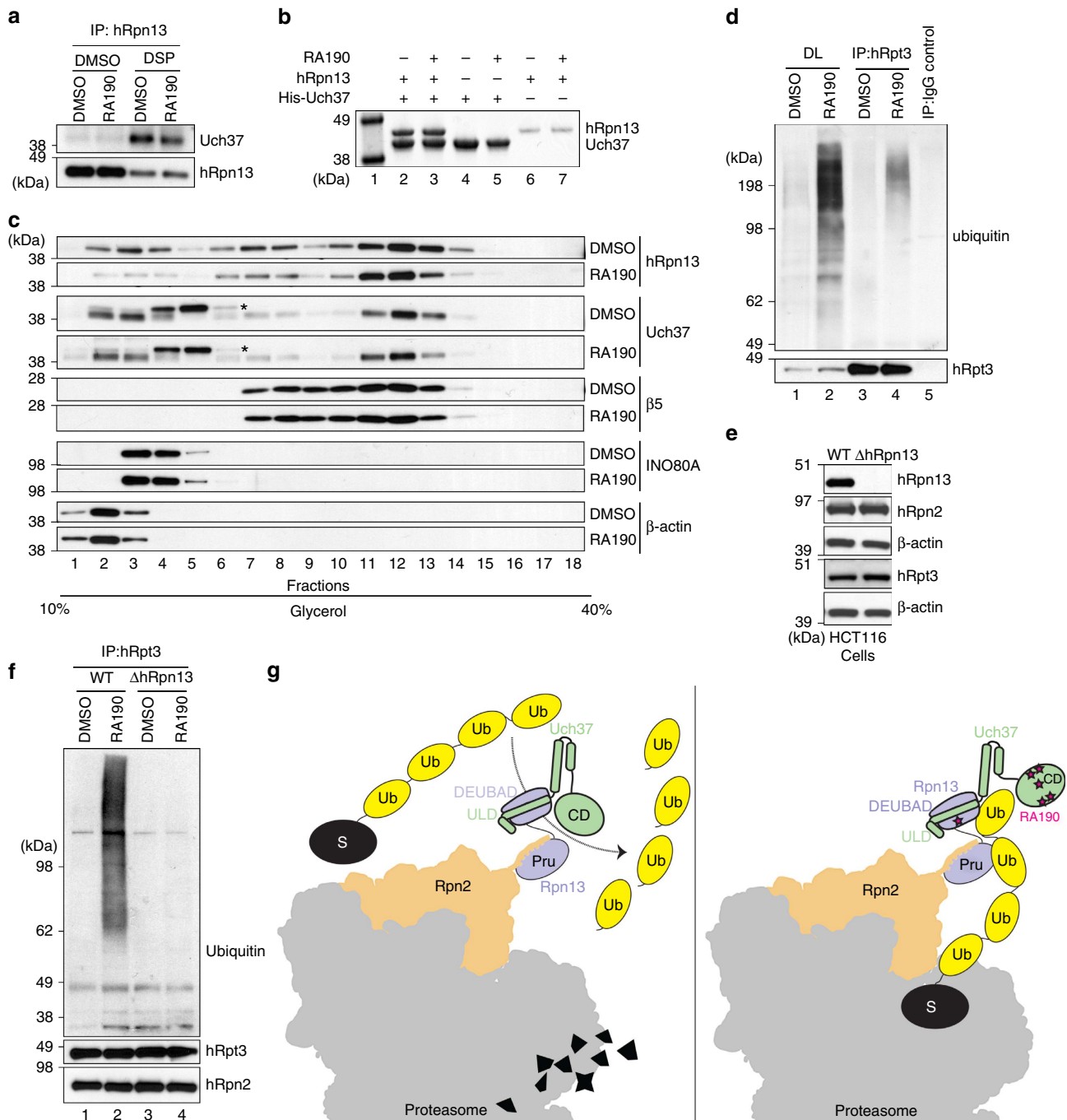

**Figure 5 | RA190 treatment leads to accumulation of ubiquitinated species at the proteasome.** (**a**) HCT116 cells were treated with 1 μM RA190 or DMSO for 24 h and then further treated with the crosslinker DSP or DMSO as a control. Cell lysates were immunoprecipitated with anti-hRpn13 antibodies and immunoprobed with anti-Uch37 antibodies. (**b**) His-Uch37, hRpn13 or equimolar mixture of these two proteins at 2 μM were incubated with 20-fold molar excess RA190 or DMSO and then Talon resin. Retention of hRpn13 on the Uch37-bound resin was assessed by SDS–PAGE with Coomassie staining. Protein markers are included in lane 1. (**c**) HCT116 cell lysates treated with 1 μM RA190 or DMSO (as a control) for 24 h were loaded onto a 10–40% linear glycerol gradient and subjected to ultracentrifugation. Gradient fractions were resolved by SDS–PAGE and immunoprobed with indicated antibodies. *Nonspecific band from the Uch37 antibody. (**d**) HCT116 cells were treated with 1 μM RA190 or DMSO control for 24 h and lysates immunoprecipitated with anti-hRpt3 antibodies, followed by immunoblotting as indicated. Direct loads (DLs) were included as well as an IgG control for the DMSO-treated cells. (**e**) Wild-type (WT) or hRpn13-deleted (ΔhRpn13) HCT116 cells were collected, lysed and immunoprobed as indicated, with β-actin as a loading control. (**f**) HCT116 WT and ΔhRpn13 cells were treated with 1 μM RA190 or DMSO control for 24 h and lysates immunoprecipitated with anti-hRpt3 antibodies, followed by immunoblotting as indicated. hRpn2 was used as a control to confirm immunoprecipitation of proteasome subunits. (**g**) Proposed mechanism of action for RA190 at the proteasome. RA190 (pink star) targets the hRpn13 DEUBAD domain without abrogating hRpn13 (periwinkle blue) interaction with ubiquitin chains (yellow) or Uch37 (green). RA190 also targets and inactivates the Uch37 catalytic domain (CD), impairing ubiquitin chain disassembly and in turn ubiquitin release at hRpn13. A portion of the 26S proteasome is represented in grey with Rpn2 in orange and substrate in black.

Uch37 seems to interfere with the ability of these functional receptors (Rpn1 and Rpn10) to clear ubiquitinated proteins from the proteasome (Fig. 5f). Future experiments are required to understand this finding mechanistically; however, it is possible that Rpn1 and Rpn10 are affected by the presence and/or occupancy of Rpn13 through allosteric relationships within the RP and/or direct interaction with Rpn13-bound substrates that, in response to Uch37 inhibition, become less dynamic or improperly oriented for deubiquitination.

Proteasomal deubiquitinating enzymes have emerged as new anticancer targets[52,59,60]. Uch37 is upregulated in multiple cancers, including epithelial ovarian cancer, hepatocellular carcinoma and oesophageal squamous cell carcinoma with high Uch37 expression associated with poor prognosis[61–63]. A compound with a similar reactive group compared with RA190 was previously reported to react with the proteasome deubiquitinating enzymes[53], in support of our finding that RA190 directly targets and inhibits Uch37.

It is possible that RA190 efficacy is achieved by two synergistic effects: targeting hRpn13 outside of the proteasome at C88 and inactivation of Uch37. All of the proteasome inhibitors currently approved for cancer therapy target the proteasome catalytic CP. A compelling aspect of this model is the synergy expected by targeting an alternative enzymatic function in the proteasome, and a recent publication indicated RA190 to be synergistic with bortezomib and carfilzomib[39]. Our findings merged with this published result suggest that inhibition of Uch37 may be effective towards restricting cancer cell proliferation with synergy towards currently FDA (Food and Drug Administration)-approved proteasome inhibitors. It is worth noting that the RA190 bis-benzylidine activity, which relies on covalent targeting of cysteine, has been identified to target many cellular proteins[53,64,65], suggesting a lack of specificity. However, loss of hRpn13, and in turn Uch37, abrogates RA190 sensitivity in HCT116 cells, as ubiquitinated proteins do not accumulate at the proteasome when hRpn13 and Uch37 are absent (Fig. 5f); thus, RA190 does not appear to affect proteasome function when hRpn13 is deleted. Indeed, specific inhibitors of Uch37 versus hRpn13 would be invaluable for further dissecting the potential of this ubiquitin receptor-deubiquitinating enzyme pair as anticancer targets. Moreover, such agents may maintain the efficacy of RA190, but with reduced off-target effects.

## Methods

**Plasmids and antibodies.** We used a p3 × FLAG-hRpn2 (916-953) expression vector[47] to produce p3 × FLAG-EGFP-hRpn2 (940–953) or p3 × FLAG-EGFP-hRpn2 (940–947), inserting enhanced green fluorescent protein (EGFP) between the 3 × FLAG tag and the hRpn2 sequence that was modified as indicated by site-directed mutagenesis (Agilent QuickChange) and validated with sequencing (Macrogen). A p3 × FLAG-EGFP expression vector was used for control experiments. Antibodies (dilutions) used in this study include anti-hRpn13 (Abcam ab157185, 1:5,000 or 1:10,000), anti-hRpt3 (Abcam ab140515, 1:1,000), anti-hRpn2 (Abcam ab2941, 1:1,000), anti-Uch37 (Abgent AM2200a, 1:5,000), anti-FLAG (Sigma-Aldrich F1804, 1:3,000 or 1:2,000), anti-ubiquitin (EMD Millipore MAB1510, 1:1,000), anti-His (Thermo Fisher Scientific MA1-21315, 1:1,000), anti-β-actin (Cell Signaling Technology 4970, 1:3,000), anti-INO80A (Abcam ab118787, 1:2,000) and anti-β5 (Enzo BML-PW 8895-0100, 1:5,000).

**Cell culture and transfection.** HCT116 cell line was purchased from the American Tissue Culture Collection (ATCC, cat. no. ATCC CCL-247) and the hRpn13-deletion (ΔhRpn13) HCT116 cell line generated in this study, as described below. Cells were grown in McCoy's 5A modified medium (ATCC), supplemented with 10% fetal bovine serum (Atlanta Biologicals) in a 37 °C humidified atmosphere of 5% $CO_2$. Plasmids were transfected using Lipofectamine LTX according to the manufacturer's instructions (Thermo Fisher Scientific).

**CRISPR-Cas9-mediated gene targeting strategy.** We amplified a 0.9 kb fragment upstream of exon 4 as a 5′-homologous arm and a 1.0 kb fragment downstream of exon 4 as a 3′-homologous arm in the *ADRM1* gene (encoding hRpn13)

by PCR using genomic DNA from HCT116 cells as a template. The primers used were 5′-AACTCGAGTGAAGGGGACCACCGTGACTCCG-3′ and 5′-TTGAATTCTTGGACCCTGCCTTGAACTTCAGC-3′ for the 0.9 kb fragment, and 5′-AAGGATCCATGCTGGCCCTGGTTCTAACGATG-3′ and 5′-TTTGCGGCCGCTCCGAAGGCACTTAGCTGCTGC-3′ for the 1.0 kb fragment. A targeting vector was constructed by sequentially subcloning the 5′-arm, the puromycin resistance gene cassette and the 3′-arm into pBluescript II to delete exon 4 of the *ADRM1* gene. Two pairs of single-guide RNA (sgRNA)-encoding DNA oligos that target 39 bp downstream and 217 bp downstream of exon 4 were designed (sgRNA-B and sgRNA-C, Supplementary Table 4). Each pair of annealed oligos was subcloned into the *Bbs*I site of pX330 (Addgene 42230) that which expresses sgRNA and Cas9 simultaneously.

**Establishment of hRpn13 deletion HCT116 cells.** The targeting vector and the pX330 plasmid encoding each sgRNA were transfected into HCT116 cells. The cells were cultured in Dulbecco's modified Eagle's medium supplemented with 10% fetal bovine serum and 4 µg ml$^{-1}$ puromycin for 10 days. Colonies were then picked, and deficiency of hRpn13 proteins screened by immunoblot analysis of cell lysates using anti-Rpn13 antibodies. We obtained three independent HCT116 clones deficient in hRpn13, one using sgRNA-B (B11) and two using sgRNA-C (C8 and C9). Clone B11 was used exclusively for this study.

**Quantitative real-time PCR.** Six independent total RNA samples from HCT116 WT or ΔhRpn13 cell lines were purified by using the RNeasy Plus Mini Kit (QIAGEN), and complementary DNA (cDNA) synthesized by using the iScript cDNA synthesis kit (1708890, Bio-Rad). hRpn13 and glyceraldehyde-3-phosphate dehydrogenase (GAPDH) mRNA expression was measured by using a real-time PCR (CFX96, Bio-Rad) instrument with specific PrimeTime TaqMan primers (hRpn13-Hs.PT.56a.79495 (PrimeTime Primer 1: 5′-GCTTGAACTCA-CAGTCGTCA-3′; PrimeTime Primer 2: 5′-GACTCGCTTATTCACTTCTGC-3′; PrimeTime Probe: 5′-ATCAAGTCGTCTTCCACGTTCCCG-3′), GAPDH-Hs.PT.42.61714 (PrimeTime Primer 1: 5′-CATGTAAACCATGTAGTTGAGGT-3′; PrimeTime Primer 2: 5′-AAGGTGAAGGTCGGAGTCA-3′; PrimeTime Probe: 5′-CGGATTTGGTCGTATTGGGCGC-3′), Integrated DNA Technologies). GAPDH was used as an internal standard. Statistical analysis was performed by a two-tailed, two-sample equal variance Student's *t*-test with P values ⩽0.05 considered significant.

**Crosslinking coupled immunoprecipitation.** HCT116 cells were washed with phosphate-buffered saline (PBS) followed by incubation with 0.5 mM DSP (Thermo Fisher Scientific) freshly prepared. After incubating at room temperature for 30 min, the reaction was quenched for 15 min with 20 mM Tris-HCl, pH 7.5. Cells were then collected, washed with PBS and lysed in RIPA buffer (Thermo Fisher Scientific) supplemented with 1 mM phenylmethylsulfonyl fluoride (Sigma-Aldrich) and a protease inhibitor cocktail (Roche). Lysates were subjected to immunoprecipitation with anti-hRpt3 or anti-hRpn13 antibodies. Before immunoblotting, 100 mM 1,4-dithiothreitol (DTT) was added to the direct loads and immunoprecipitated protein complexes.

**Immunoprecipitation.** HCT116 WT or hRpn13 deletion (ΔhRpn13) cells were collected and washed with PBS followed by lysing in either RIPA or 1% Triton-TBS buffer (50 mM Tris-HCl, pH 7.5, 150 mM NaCl, 1 mM EDTA) supplemented with protease inhibitor cocktail (Roche). Total protein concentration was determined by bicinchoninic acid (Pierce) or Bradford (Sigma-Aldrich). Lysates were precleared with protein G sepharose (Sigma-Aldrich) for 1 h, incubated with the indicated antibodies overnight at 4 °C and then incubated for an additional 3 h at 4 °C with protein G sepharose. A control experiment was done with rabbit IgG and the DMSO-treated cell lysates. Following extensive washing, proteins bound to the protein G sepharose were eluted and analysed by immunoblotting.

**Immunoblotting.** Cells were lysed and protein concentration determined as described above. Protein lysates were loaded onto 4–12% Bis-Tris polyacrylamide gels (Life Technologies), subjected to SDS–PAGE and transferred to Invitrolon polyvinylidene difluoride membranes (Life Technologies). The membranes were blocked in Tris-buffered saline with 0.1% Tween-20 (TBST) supplemented with 5% skim milk, incubated with primary antibody, washed in TBST, incubated in secondary antibody and finally washed extensively in TBST. Amersham ECL regular or enhanced chemiluminescent reagent (GE Life Sciences) was used for antibody signal detection. Uncropped images for Figs 1c,3g,h,4a and 5a–f are included in the Supplementary Fig. 7.

**Immunoblot quantification and statistical analyses.** After immunoblotting, the films were scanned and protein band density quantified using ImageJ. Protein abundance was normalized to the signal of immunoprecipitated protein and Excel used to perform the statistical analyses. A two-tailed, two-sample equal variance Student's *t*-test was performed with P values ⩽0.05 considered significant.

**Glycerol gradient centrifugation and fractionation.** Glycerol gradients were made in a buffer containing 50 mM Tris-HCl, 1 mM DTT, 1 mM ATP, 5 mM MgCl$_2$, pH 7.6 in a volume of 1.7 ml. Cells were treated with 1 μM RA190 or DMSO as a control for 24 h and whole-cell lysates from HCT116 cells were fractionated by ultracentrifugation with a 10–40% glycerol gradient (Beckman Coulter, SW41 Ti rotor, 131,600 g for 18 h at 4 °C). Gradient fractions were collected and resolved by SDS–PAGE (4–12% Bis-Tris gel) and immunoprobed with indicated antibodies with dilutions mentioned in the above section.

**ITC experiments.** ITC experiments were performed at 25 °C on a MicroCal iTC200 system (Malvern, PA, USA). hRpn13 Pru and hRpn2 (940–953) samples were dialysed extensively against ITC buffer (20 mM sodium phosphate, 50 mM NaCl and 10 mM βME, pH 7.0) while hRpn2 (944–953) was dissolved in ITC buffer. One aliquot of 0.5 μl followed by 18 aliquots of 2.1 μl of 200 μM hRpn2 (940–953) or hRpn2 (944–953) were injected at 1,000 r.p.m. into the calorimeter cell (volume 200.7 μl) that contained 20 μM or 18 μM hRpn13 Pru, respectively. Blank experiments were performed by replacing protein samples with buffer and the resulting data subtracted from the experimental data during analyses. The integrated interaction heat values were normalized as a function of protein concentration, and the data were fit with MicroCal Origin 7.0 software. Binding was assumed to be at one site to yield the binding affinity $K_a$ (1/$K_d$), stoichiometry and other thermodynamic parameters. The peptide hRpn2 (944–953) was synthesized on a Liberty Blue Microwave peptide synthesizer (CEM Corporation) using Fmoc chemistry on Wang resin, cleaved from the resin and purified by high-performance liquid chromatography (HPLC) in a manner similar to that described below for the synthesis of FITC-hRpn2 (940–953).

**Differential scanning fluorimetry.** Thermal stability of the protein and complexes was measured using label-free DSF on a Prometheus NT.48 instrument (Nano-Temper Technologies, Germany). The shift of intrinsic tryptophan fluorescence upon temperature-induced unfolding was monitored measuring emission fluorescence at 350 nm. The samples were loaded in standard glass capillaries (Nano-Temper) and subjected to heating from 20 to 85 °C at a rate of 1 °C per min. Melting temperatures were obtained from three independent scans and calculated from the first derivative of the tryptophan emission intensities at 350 nm.

**Synthesis of FITC-labelled hRpn2 probe for FP assays.** The hRpn2 sequence (QEPEPPEPFEYIDD) was synthesized using automated, microwave-assisted solid-phase peptide chemistry with Fmoc-protection on a CEM Liberty peptide synthesizer on Wang resin. The N-terminus was extended by using commercial {2-[2-(Fmoc-amino)ethoxy]ethoxy} acetic acid (Chem-Impex Int cat. no. 07310) and capped with 4-pentynoic acid (Chem-Impex cat. no. 26911). The final peptide was cleaved from the resin in trifluoroacetic acid (TFA)/triisopropylsilane/water (95/2.5/2.5) and purified by preparative HPLC (gradient elution, 100% H$_2$O to 40/60 water/MeCN with 0.1% TFA added). Fluorescein isothiocyanate, isomer I (FITC) (39 mg, 0.10 mmol) was dissolved in anhydrous N,N-dimethylformamide (1 ml) with magnetic stirring in a round-bottom flask. N,N-diisopropylethylamine (35 μl, 0.20 mmol) was added to the solution, followed by 3-azidopropylamine (10 mg, 0.1 mmol). The reaction was allowed to stir overnight, then concentrated to dryness under vacuum. The resulting residue was purified directly using combiflash (Telodyne Isco CombiFlash 200i, gradient elution of 0–10% methanol in dichloromethane). Fractions containing pure product were combined and concentrated to provide FITC-N$_3$ (40 mg, 0.08 mmol, 82% yield) as a dark orange solid. FITC-N$_3$ was then coupled to the hRpn2-alkyne using copper-mediated click chemistry. Briefly, 1 equivalent each of FITC-N$_3$ and hRpn2-alkyne were mixed in 1:1 DMSO/water with CuSO$_4$ (25 mol% from a 4% w/v in H$_2$O), tris(3-hydroxypropyltriazolylmethyl)amine (THPTA, 50 mol% from a 100 mM DMSO stock) and sodium ascorbate (5 equivalent from a 0.5 M aqueous solution). The click reaction proceeded overnight at room temperature, and the product purified directly by preparative HPLC (gradient elution, 90/10 water/MeCN to 100% MeCN with 0.1% TFA added) to >95% purity.

**FP assays.** The assay buffer used to determine the affinity of hRpn13 for FITC-hRpn2 (940–953) was 20 mM sodium phosphate, 50 mM NaCl, pH 6.5. For $K_d$ determination of full-length hRpn13 with FITC-hRpn2 (940–953), 30 μl of serially diluted hRpn13 or assay buffer was added to the wells of a 384-well plate. Then, 10 μl of 40 nM FITC-hRpn2 solution was added to each well containing hRpn13 protein or buffer control. For inhibition experiments with RA190, assay buffer contained 2% DMSO. The plate was incubated at room temperature for 30 min with shaking, then FP was read using a BioTek Synergy 2 plate reader (excitation 485/20 nm, emission 528/20 nm). The experiment was performed in triplicate and FP values with the corresponding value of probe alone subtracted plotted against hRpn13 concentration. Analyses were performed by using nonlinear regression in GraphPad Prism 7 with the data fit as specific binding to a Hill slope model. The indicated $K_d$ value represents average ± s.d.

To determine the effect of RA190 on binding affinity, 20 μl of serially diluted hRpn13 or assay buffer was added to the wells of a 384-well plate containing 10 μl

of 400 μM RA190 (8% DMSO) or buffer with 8% DMSO. Following 30 min of preincubation at room temperature, 10 μl of 40 nM FITC-hRpn2 solution was added to each well. Final concentrations were 100 μM RA190, 10 nM FITC-hRpn2 and 2% DMSO. The plate was incubated at room temperature for 30 min with shaking, then FP was read and the data analysed as described above.

The competition assay with RA190 was performed in triplicate with 10 μl of serially diluted RA190 in assay buffer containing 8% DMSO added to the wells of a 384-well plate, along with DMSO controls. Then, 20 μl of 60 nM Pru domain was added to the wells containing serially diluted RA190 and incubated for 30 min at room temperature with shaking. Next, 10 μl of 40 nM FITC-hRpn2 was added for a final concentration of 30 nM hRpn13 Pru domain, 10 nM FITC-hRpn2 and 2% DMSO. The plate was incubated for an additional 30 min at room temperature with shaking and FP read as described above. FP values were background subtracted (probe alone), plotted versus RA190 concentration, and analysed by nonlinear regression in GraphPad Prism 7 (data were fit using the log(inhibitor) versus response–variable, four parameter model).

**LC-MS analyses of RA190 adducts.** hRpn13 Pru or hRpn2 (940–953) was purified as described below. hRpn13 DEUBAD (253–407) was purified in a similar manner to hRpn2 (940–953), but with a thrombin cleavage site. Preparation of His-Uch37 was similar to hRpn13 Pru, but with elution from Talon Metal Affinity resin in imidazole-containing buffer (20 mM sodium phosphate, 300 mM NaCl, 250 mM imidazole, 2 mM DTT, pH 6.5) and the His tag retained. The purified proteins were dialysed extensively against phosphate buffer (20 mM sodium phosphate, 50 mM NaCl, pH 6.5) to remove DTT and complexes formed as indicated by incubating sample mixtures on ice for >1 h. Next, 20 or 50 μM RA190 (Xcessbio, San Diego, CA, USA) was reacted with 100 μl of 2 μM target sample by incubation at 4 °C for 2 h while rotating. To test for reversibility, hRpn13 Pru was incubated with 10-fold molar excess RA190 for 1 h at 4 °C, and then 10-fold molar excess hRpn2 (940–953) or equivalent volume of buffer (as a control) for another 1 or 19 h at 4 °C for a final concentration of 2 μM hRpn13 Pru and 20 μM RA190 with or without 20 μM hRpn2 (940–953). For LC-MS analysis, acetonitrile was added to RA190-treated samples to a final concentration of 10%. LC-MS was performed on either an Agilent (Agilent Technologies, Inc., Santa Clara, CA, USA) 6100 Series Quadrupol LC/MS System or 6520 Accurate-Mass Q-TOF LC/MS System, each equipped with a dual electro-spray source, operated in the positive-ion mode. Data acquisition and analysis were performed by OpenLAB CDS ChemStation Edition C.01.05 or Mass Hunter Workstation (version B.06.01). For data analysis and deconvolution of mass spectra, Mass Hunter Qualitative Analysis software (version B.07.00) with Bioconfirm Workflow was used.

**In vitro deubiquitination assay.** Purified His-Uch37 and hRpn13 were dialysed extensively against phosphate buffer to remove DTT and K48-linked tetraubiquitin (Boston Biochem) was dissolved in the same buffer. Uch37, hRpn13 or hRpn13-Uch37 were incubated with RA190 or DMSO rotating at 4 °C for 2 h, followed by addition of K48-linked tetraubiquitin (final concentration of 1 μM K48-linked tetraubiquitin, 0.1% DMSO in all of reactions and final concentrations of 1 μM Uch37, hRpn13, hRpn13-Uch37 or 20 μM RA190) for another 8 h at 37 °C. The reaction was quenched by adding SDS–PAGE loading buffer (2% SDS, 10% glycerol, 2 M urea, 0.01% Bromophenol Blue, 200 mM DTT) and heating at 80 °C for 8 min. Samples were subjected to SDS–PAGE and immunoblot analysis.

**Pulldown assay.** hRpn13 and His-Uch37 were dialysed extensively against buffer (20 mM sodium phosphate, 50 mM NaCl, pH 6.5) to remove DTT. His-Uch37, hRpn13 or an equimolar mixture of these two proteins at 2 μM concentration were separately incubated with 20-fold molar excess RA190 or 0.2% DMSO (RA190 is dissolved in DMSO) for 2 h at 4 °C, followed by 25 μl of Talon resin (Clontech) for 40 min. The resin was then washed three times with buffer maintaining the same amount of RA190 or DMSO. Resin-bound proteins were next denatured by addition of SDS–PAGE loading buffer and heating at 80 °C for 8 min, and then visualized by SDS–PAGE with Coomassie blue staining.

**NMR sample preparation.** hRpn2 (940-953) or hRpn13 Pru (150) was expressed in Escherichia coli BL21(DE3) pLysS (Invitrogen) as a recombinant protein fused with glutathione S-transferase or His tags at the N-terminus followed by a Pre-Scission protease cleavage site. Cells were grown at 37 °C to an OD value of 0.6 and isopropyl-β-D-thiogalactoside (0.4 mM) to induce protein expression for 4 h at 37 °C or 20 h at 17 °C. The cells were collected by centrifugation at 4,550 g for 30 min, lysed by sonication and cell debris removed by centrifugation at 31,000 g for 30 min. The lysates were incubated with Glutathione S-sepharose 4B (GE Healthcare Life Sciences) for 3 h or Talon Metal Affinity resin (Clontech) for 1 h, and the resin washed extensively with buffer (20 mM sodium phosphate, 300 mM NaCl, 2 mM DTT, pH 6.5). hRpn2 (940–953) or hRpn13 Pru (1–150) was eluted from the resin by overnight incubation with 50 units per ml PreScission protease (GE Healthcare Life Sciences) in buffer (20 mM sodium phosphate, 50 mM NaCl, 2 mM DTT, pH 6.5). The eluent was subjected to size exclusion chromatography with a Superdex75 column on an FPLC system for further purification. A mixture of 1.5-molar excess hRpn2 (940-953) with hRpn13 Pru was prepared from the

separately purified proteins and then passed over the Superdex75 column again. $^{15}N$ ammonium chloride and $^{13}C$ glucose were used for isotopic labelling. All NMR experiments were performed in phosphate buffer that included 2 mM DTT, 0.1% sodium azide and 10% $D_2O$ or 100% $D_2O$.

**NMR experiments.** All NMR experiments were conducted at 25 °C on Bruker Avance 600, 700, 800 or 850 MHz spectrometers equipped with cryogenically cooled probes. $^1H$, $^{15}N$, $^{13}C$ HNCACO, HNCO, HNCACB and CBCACONH spectra were acquired on a mixture of 0.7 mM $^{15}N$-, $^{13}C$-labelled hRpn13 Pru and equimolar unlabelled hRpn2 (940–953). Distance constraints for structure calculations were obtained by using an $^{15}N$-dispersed NOESY spectrum recorded on a mixture of 0.6 mM $^{15}N$-, $^{13}C$-labelled hRpn13 Pru and equimolar $^{15}N$-, $^{13}C$-labelled hRpn2 (940-953) with a 150 ms mixing time as well as $^{13}C$-edited NOESY spectra on mixtures of 0.7 mM $^{15}N$-, $^{13}C$-labelled hRpn13 Pru and equimolar unlabelled hRpn2 (940–953) (120 ms mixing time) or 0.7 mM $^{15}N$-, $^{13}C$-labelled hRpn2 (940–953) and equimolar unlabelled hRpn13 Pru (100 ms mixing time). Intermolecular NOE distance constraints were determined by using a $^{13}C$-half-filtered NOESY spectrum (100 ms mixing time) recorded on a mixture of 0.7 mM $^{15}N$-, $^{13}C$-labelled hRpn13 Pru and equimolar unlabelled hRpn2 (940-953). The $^{13}C$-edited NOESY spectra were acquired on samples dissolved in $D_2O$. NMRPipe[66] was used to process data and XEASY[67] was used to visualize and analyse spectra.

**Structure determination.** The 179 backbone $\varphi$ and $\psi$ torsion angle constraints were generated by TALOS+ (http://spin.niddk.nih.gov/bax/software/TALOS/) based on HN, C$\alpha$, C$\beta$, CO and N chemical shift assignments. NOE interactions were used in combination with secondary structure information by TALOS+ to define 45 intramolecular hydrogen bonds for hRpn13 Pru in the hRpn2-bound state. One intermolecular hydrogen bond was determined by NOE interactions between hRpn13 V38 HN and hRpn2 F948 HN in a $^{15}N$-dispersed NOESY spectrum recorded on a mixture of 0.6 mM $^{15}N$-, $^{13}C$-labelled hRpn13 Pru and equimolar $^{15}N$-, $^{13}C$- labelled hRpn2 (940-953) with a 150 ms mixing time. Distances for hydrogen bonds were set between the acceptor oxygen and donor hydrogen and nitrogen of 1.8–2.1 Å and 2.5–2.9 Å, respectively. These constraints were combined with 1,782 intramolecular and 140 intermolecular NOE-derived distance constraints (Table 2) to calculate the structure of the hRpn13 Pru-hRpn2 (940-953) complex by using simulated annealing algorithms in XPLOR-NIH 2.33 (http://nmr.cit.nih.gov/xplor-nih/). Briefly, 20 linear starting structures were subjected to 19,400 simulated annealing and cooling steps of 0.005 ps. The lowest energy structure with best geometry was then used as the starting structure for a second iteration of simulated annealing to generate 200 structures. The 12 lowest energy structures without distance or dihedral angle violations >0.5 Å or 5° respectively were finally selected for visualization and statistical analyses. Structure evaluation was performed with the program PROCHECK-NMR[68]; the percentage of residues in the most favoured, additionally allowed, generously allowed and disallowed regions is 84.0, 13.7, 2.3 and 0.0, respectively. Visualization was performed with MOLMOL[69], PyMOL (PyMOL Molecular Graphics System, http://www.pymol.org) and CCP4mg[70].

**Data availability.** The structural coordinates and chemical shift data for hRpn13 Pru-hRpn2 (940–953) have been deposited into the Protein Data Bank (PDB) and Biological Magnetic Resonance Data Bank (BMRB) with respective accession codes 2NBK and 25979. The UniProt accession codes Q99460 (hRpn2), Q16186 (hRpn13) and Q9Y5K5 (Uch37), PDB accession codes 4CR2 and 5IRS and EMDB accession code EMD-2594 were used in this study. All other data are available from the corresponding author on reasonable request.

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

## Acknowledgements

This research was funded by the Intramural Research Program of the NIH, National Cancer Institute, Center for Cancer Research. We are grateful to Xiang Chen and Fahu He for technical assistance and to Hiroshi Matsuo and Gwen Buel for critical reading.

## Author contributions

The project was conceived of by K.J.W. and X.L.; X.L. purified recombinant proteins, performed NMR experiments, solved the hRpn13 Pru-hRpn2 (940-953) complex structure and performed Uch37 functional assays. ITC was done by X.L. and S.G.T.; LC-MS was acquired and analysed by X.L. and M.D.; U.N., F.L. and V.S. performed the cell biology experiments; reagents were generated by L.R.; X.Z. synthesized FITC-hRpn2 (940-953) and D.H. performed the FP assay; N.I.T. synthesized peptides for and performed the DSF experiment. J.H. and S.M. generated the hRpn13-deleted HCT116 cell line. K.J.W. and X.L. wrote the manuscript with contribution from all authors.
