## [Peer review file · Nature Communications]

Reviewers' comments:

Reviewer #1 (Remarks to the Author):

This paper reports biochemical and biophysical experiments directed towards better understanding how the ubiquitin receptor Rpn13 interacts with the proteasome and the deubiquitylase Uch37. This is important, as recent work from various laboratories has implicated Rpn13 as an attractive target for chemotherapy. The authors provide structural detail to the previous cartoon-level model of Rpn13 binding to a C-terminal domain of the Rpn2 protein of the proteasome. These data define the exact residues of Rpn2 that contact Rpn13 and will be of great interest to people interested in developing inhibitors of this protein-protein interaction.

Perhaps the most important data reported in this paper has to do with the mechanism by which a small molecule called RA-190 inhibits the proteasome. This is also a topic of great interest, since RA-190 exhibits interesting activity against various cancers, but with reduced toxicity compared to classical active site-targeted proteasome inhibitors such as Bortezomib. Previous work claimed that this chalcone-containing compound is a covalent inhibitor of Rpn13 with a high level of selectivity for Cys88 of Rpn13. The data in this paper strongly refute this model. The authors find that RA-190 cannot access Cys88 when bound to Rpn2. However, they present convincing mass spectrometry-based evidence that RA-190 is a promiscuous alkylators of other cysteines, which should be a surprise to no one. In particular, the data presented in this paper show that RA-190 directly inactivates Uch37, the deubiquitylase that associate with Rpn13 and is activated by it. This argues strongly for a mechanism of action of RA-190 quite different than the one accepted in the literature currently. Thus, the data in this paper will be quite interesting to people interested in alternative approaches to drug the proteasome.

In general, this paper deserves to be published in Nature Communications after a few minor modifications.

The only significant point is that the authors should comment on a possible mechanism of action of RA-190 in which it alkylates newly synthesized Rpn13 and prevents it from association with the proteasome. This is the only model that I can see that would accommodate the authors' biophysical data yet accommodate an Rpn13-targeted mechanism of action. Along these lines, additional experiment that would be useful is to alkylate Rpn13 with RA-190 and demonstrate directly that it fails to bind to the Rpn2 C-terminal peptide. I do not recall seeing this in the manuscript.

A small thing is that there are a few awkward sentences in the paper.

Line 196: "...as described in38.

Lines 294-5: "Herein we report an impressive binding mechanism..."

Line 303: Although this experiment implements hRpn13 C88... (I believe they meant "implicates")

These should all be cleaned up by a careful re-read of the manuscript.

Finally, since RA-190 is a covalent inhibitor and its activity is time-dependent, the authors should be careful to point out how long they treated the proteins and with what concentration of RA-190 to obtain the alkylation patterns that they observe. They should make clear if this is analogous to what was used in previous studies with RA-190. They state things in terms of equivalents of drug to protein, but that is not the critical issue here .

Reviewer #2 (Remarks to the Author):

This manuscript reports on the NMR structure of the hRpn13 proteasome ubiquitin receptor (the Pleckstrin homology, or Pru domain) bound to hRpn2 residues 940-953. This tight interaction serves to dock the Rpn13 receptor to the proteasome. The authors determined that the minimal Rpn2 fragment binds with high, nanomolar affinity (using ITC), and that the complex is thermodynamically more stable than the hRpn13 Pru domain alone. Overall, the NMR structural studies and calorimetry experiments are technically sound. The Pru domain from Rpn13 is poorly defined in the latest cryoEM structures for the intact proteasome. The authors dock the structure of the complex into a cryoEM reconstruction to show that the N-terminal residues form the Pru are unstructured and may be responsible for density observed at Asn20. However, the spherical structure of the Pru and the resolution of the reconstruction cannot allow for a definitive answer regarding position within the proteasome, though this may not be important as flexibility of the Pru domain may be required for substrate recognition.

The structure of the Rpn13/Rpn2 complex raises questions regarding a previously proposed mechanism of action for the small molecule covalent modifier RA190. This molecule is believed to react with Cys88 within the Pru domain of Rpn13 and inhibit the proteasome. Based on this study, Rpn2 binding to Rpn13 will occlude the RA190 site, and the high affinity will inhibit reaction of RA190 with Cys88. Using mass spectrometry, the authors showed convincingly that RA190 reacts only with the single Cys within the DEUBAD (deubiquitinase adaptor domain) in the presence of Rpn2. Immunoprecipitation experiments with antibodies against a proteasome ATPase member indicate that close association with proteasome components shields the Pru domain from RA190 but not the DEUBAD domain.

The reactivity of Rpn13 with RA190 in the presence of Rpn2 will be dependent upon reaction conditions, and left long enough, and at higher temperatures, RA190 will eventually react with all the Cys residues on Rpn13. However, the in vitro data clearly indicate that Rpn2 is capable of inhibiting the reaction, and the cell studies are consistent with this as well.

Previous studies have shown that RA190 can inhibit the deubiquitinase Uch37, an enzyme that interacts with Rpn13 and hydrolyzes K48 ubiquitin links. Using K48-Ub4 chains, the authors showed that Uch37 primarily hydrolyzes Ub4 to Ub3 plus Ub1. As previously shown, Rpn13 significantly accelerates the reaction. The control experiments for these assays clearly indicate that RA190 inhibits Uch37 in the presence and absence of Rpn13, implicating Uch37 as an in vivo target. Mass spec results indicate that RA190 reacts with every Cys residue in Uch37. For these assays, the rate of Ub4 hydrolysis seems slow in comparison to the k_{cat} values for Ub-AMC hydrolysis reported by Hill (Mol Cell 57). At the concentration of Uch37 employed, it is possible that Uch37 is subject to a dimerization dependent inhibition.

The experimental data suggest that inactivation of Uch37 by RA190 in cells should lead to build up of ubiquitinated proteins. In support of this, RA190 treatment of HCT116 cells leads to increased levels of ubiquitinated proteins, as determined by co-IP experiments with proteasome ATPase components. This is an important result, as the enzymes of the ubiquitin conjugation machinery that attach Ub to proteins depend on Cys residues, implying that reactivity of RA190 with Uch37 plays a role in proteasome inhibition in HCT116 cells.

Overall the work is well done, with appropriate controls. The proteasome is a major target for drug development, therefore, the manuscript is likely to be of broad interest. Questions that remain involve the selectivity and specificity of RA190 for Uch37, and potential off-target effects in cells. Also, the authors stop short of implicating the active site Cys88 of Uch37 as the target of RA190. Is the effect of RA190 due to covalent modification of this residue?

Reviewer #3 (Remarks to the Author):

The submitted manuscript entitled “Structure of the Rpn13:Rpn2 complex leads to new insights for Rpn13 and Uch37 as anti-cancer targets” presents the NMR structure of the ubiquitin-binding domain of the proteasomal ubiquitin receptor hRpn13 with a short fragment of hRpn2, which serves as its proteasome docking site (940-953).

The same group has previously demonstrated that a bit longer, 38-amino acid peptide derived from the C-terminal end of hRpn2 (916-953) binds to hRpn13 with 12 nM Kd. By using NMR, they have also previously identified the hRpn13-interacting amino acids in hRpn2 fragment (PLoS One 2015; 10(10): e0140518). Now they identified a smaller fragment of hRpn2 (940-953) sufficient for hRpn13 binding and determined dissociation constant (Kd) of 27 nM for binding with hRpn13 Pru.

On a collaborative basis, the group of Kylie Walters has previously described covalent binding of bis-benzylidene piperidone RA190 to (as suggested) Cys88 residue of Rpn13, which is located within its ubiquitin-binding domain Pru. RA190 binding to Rpn13 Pru domain was shown to affect proteasomal function and to lead to the rapid accumulation of polyubiquitinated proteins (Cancer Cell 2013; 24(6): 791-805).

In the new manuscript, however, they present mass spectrometry analyses indicating that RA190 does not target hRpn13 Pru when bound to hRpn2, consistent with their hRpn13:hRpn2 complex structure.

Interestingly, in the initial paper they identified 42 kDa 19S RP proteasomal protein as RA190 target and actually excluded UCH37 (as well as Rpn10 and HHR23B) as RA190 target (Cancer Cell 2013; 24(6): 791-805).

In the new manuscript, however, they present data on RA190 binding to the hRpn13 Uch37-binding DEUBAD domain and Uch37 itself, which in turn inactivates the DUB catalytic activity.

Questions

1. New data suggest that basically any Cys residue in UCH37 can be targeted by UCH37, questioning the specificity of the compound. What about other proteasomal DUBs (USP14 and Rpn11) – are they also modified by RA190? What about non-proteasomal targets? Since UCH37 functions in both 26S proteasome and INO80 chromatin remodeler complexes, is nuclear UCH37 also targeted by RA190? Mass spectrometry data should be provided.
2. Figure 3G. Does amount of proteasome-bound UCH37 change upon Rpt3 IP in the presence of RA190? Is it consistent with data in Figures 5A and 5B? How about levels of 20S subunits? What is the functional consequence of RA190 binding to DEUBAD domain?
3. Kd for hRpn2:hRpn13 Pru binding differs when hRpn2 (940-953) or hRpn2 (916-953) are used. Since binding is stronger with the larger fragment, it seems that additional residues contribute to the binding in the larger hRpn2 fragment. It would be beneficial to explain/model/determine it.
4. I have a conceptual problem dealing with the fact that 2 previous publications by the same group have basically shown the contradictory data than data presented in this manuscript. I think that the authors should provide more elaborate explanation why the previous results

were obtained, further strengthening the correctness of the current finding.

5. There might be alternative explanations why hRpn2-bound hRpn13 does not bind RA190, which cannot be excluded, including the potential extra-proteasomal/non-Rpn2-bound fraction of Rpn13 binding to RA190. I suggest to include NMR or similar type of data to further study the effect of increasing amounts of RA190 on hRpn2:hRpn13 interaction, in order to further strengthen Figure 3C-D data.

Minor comments

English should be corrected (see few examples below)

28: a narrow canyon of the ubiquitin-binding hRpn13 Pru domain. Interesting (!), mass

31: hRpn13 Uch37-binding DEUBAD domain and Uch37, rendering it catalytically inactivate (!).

In summary:

Provides strong evidence for its conclusions: yes

Novel: reasonably novel, follow up of previous studies, with some of which it contradicts

Of extreme importance to scientists in the specific field: medium importance

Interesting to researchers in other related disciplines: medium interesting

Response to Reviewers

Reviewer 1

In general, this paper deserves to be published in Nature Communications after a few minor modifications.

Response: We are highly grateful to this Reviewer for their critical reading of our manuscript as well as their detailed summary culminating in the sentence above. We have addressed all of the points raised by this Reviewer below and implemented all suggestions. The manuscript is certainly improved and we thank this reviewer.

The only significant point is that the authors should comment on a possible mechanism of action of RA-190 in which it alkylates newly synthesized Rpn13 and prevents it from association with the proteasome. This is the only model that I can see that would accommodate the authors' biophysical data yet accommodate an Rpn13-targeted mechanism of action. Along these lines, additional experiment that would be useful is to alkylate Rpn13 with RA-190 and demonstrate directly that it fails to bind to the Rpn2 C-terminal peptide. I do not recall seeing this in the manuscript.

Response: We understand that this comment is meant to address why Rpn13 is required for RA190 sensitivity and that the Reviewer is kindly providing a possible mechanism of action.

We tested whether hRpn13 alkylated by RA190 fails to bind hRpn2 (940-953) by fluorescence polarization, but no inhibition was observed even in the presence of 8000-fold molar excess RA190; we have now included this data in Figure 3g of the revised manuscript. In addition, we do not observe less hRpn13 at proteasomes of RA190-treated cells (Fig. 3h and Fig. 5c). Rather, we find that RA190 is readily displaced by hRpn2 (940-953), data that is included in the revised manuscript as Supplementary Figure 4b.

It is possible that in addition to its function at the proteasome, hRpn13 performs functions outside of the proteasome. We have now included a fractionation experiment indicating the presence of hRpn13 outside of the proteasome (**Fig. 5c**), as well as a discussion of this point that highlights the finding that hRpn13 knockdown by siRNA leads to reduced Uch37 protein levels^{1,2}. *[Text redacted]*

[figure redacted]

1. Hamazaki, J. et al. A novel proteasome interacting protein recruits the deubiquitinating enzyme UCH37 to 26S proteasomes. *EMBO J* **25**, 4524-36 (2006).
2. Randles, L., Anchoori, R.K., Roden, R.B. & Walters, K.J. Proteasome Ubiquitin Receptor hRpn13 and its Interacting Deubiquitinating Enzyme Uch37 are Required for Proper Cell Cycle Progression. *J Biol Chem* (2016).

A small thing is that there are a few awkward sentences in the paper.
Line 196: "...as described in³⁸."

Response: We have now replaced the above with "as described previously³⁸."

Lines 294-5: "Herein we report an impressive binding mechanism..."

Response: We have removed this text in our revision of the Discussion section.

Line 303: Although this experiment implements hRpn13 C88... (I believe they meant "implicates")

Response: Thank you for catching this!

Finally, since RA-190 is a covalent inhibitor and its activity is time-dependent, the authors should be careful to point out how long they treated the proteins and with what concentration of RA-190 to obtain the alkylation patterns that they observe. They should make clear if this is analogous to what was used in previous studies with RA-190. They state things in terms of equivalents of drug to protein, but that is not the critical issue here.

Response: Thank you for this suggestion. We have now clarified in the text, figure captions and methods our procedures for RA190 treatment.

Reviewer 2

We are highly grateful to this Reviewer for their critical reading of our manuscript as well as their detailed summary. We have addressed the comments in the critique and added to the Discussion a clarification as suggested by the Reviewer.

The authors dock the structure of the complex into a cryoEM reconstruction to show that the N-terminal residues from the Pru are unstructured and may be responsible for density observed at Asn20. However, the spherical structure of the Pru and the resolution of the reconstruction cannot allow for a definitive answer regarding position within the proteasome, though this may not be important as flexibility of the Pru domain may be required for substrate recognition.

Response: We agree with the points made by this Reviewer and have better clarified the text corresponding to this part of the manuscript as shown below.

The resolution in the Rpn13 region of the reconstruction is low; nonetheless, by fusing the hRpn13-binding region of hRpn2 to the appropriate site in scRpn2, a favored orientation is suggested for Rpn13 in the density map (Fig. 1g). It is worth noting that the hinge between the Rpn2 region that binds hRpn13 and the preceding toroidal PC-repeat domain is undoubtedly flexible. This flexibility would provide conformational freedom for the hRpn2-bound hRpn13 Pru domain, facilitating capture of substrates.

Overall the work is well done, with appropriate controls. The proteasome is a major target for drug development, therefore, the manuscript is likely to be of broad interest. Questions that remain involve the selectivity and specificity of RA190 for Uch37, and potential off-target effects in cells.

Response: We agree with the points made by this Reviewer and in the text articulate more clearly that RA190 is not likely to react only with hRpn13 and Uch37. We also used a HCT116

cell line in which hRpn13 is deleted to assay whether hRpn13 and Uch37 are required at the proteasome for RA190-induced accumulation of ubiquitinated proteins. We find that ubiquitinated proteins do not accumulate at the proteasome in HCT116 cells with hRpn13 deleted; this data is now included as Fig. 5f in the revised manuscript. Therefore, in the context of the proteasome we believe that RA190 is in fact specific for hRpn13 and Uch37. These points are highlighted in the new Discussion text included below.

It is worth noting that the RA190 bis-benzylidene activity, which relies on covalent targeting of cysteine, has been identified to target many cellular proteins^{53,64,65}, suggesting a lack of specificity. However, loss of hRpn13, and in turn Uch37 from the proteasome, abrogated RA190-sensitivity in HCT116 cells (Fig. 5f). Indeed, specific inhibitors of Uch37 versus hRpn13 would be invaluable for further dissecting the potential of this ubiquitin receptor-deubiquitinating enzyme pair as anti-cancer targets. Moreover, such agents may maintain the efficacy of RA190, but with reduced off target effects.

Also, the authors stop short of implicating the active site Cys88 of Uch37 as the target of RA190. Is the effect of RA190 due to covalent modification of this residue?

Response: This aspect is difficult to dissect as Uch37 has five cysteines in its catalytic domain, including one (Cys9) that is proximal to Cys88, another (Cys100) located at the opposite end of the helix that contains Cys88 (please see figure below). Thus, direct targeting of Cys88 is most likely sufficient but not necessary for RA190 inhibition. RA190 adduction to Cys9 may block the active site, and RA190-conjugated to other cysteines may cause allosteric inhibition.

Structure of Uch37:hRpn13 DEUBAD:ubiquitin-propargyl (PDB 4UEL). Complex structure of Uch37 (pale green), hRpn13 DEUBAD (periwinkle blue), and ubiquitin-propargyl (yellow) with the five cysteines in Uch37 shown in magenta.

Reviewer 3:

Thank you for your critical reading of this manuscript and for your invaluable suggestions.

Questions

1. New data suggest that basically any Cys residue in UCH37 can be targeted by UCH37, questioning the specificity of the compound. What about other proteasomal DUBs (USP14 and Rpn11) – are they also modified by RA190? What about non-proteasomal targets? Since UCH37 functions in both 26S proteasome and INO80 chromatin remodeler complexes, is nuclear UCH37 also targeted by RA190? Mass spectrometry data should be provided.

Response: This comment relates to that made above by Reviewer 2. Indeed, we do not expect RA190 bis-benzylidene activity to be selective and we have emphasized this point in the

Discussion section, as indicated in our response above to Reviewer 2. However, as mentioned above we also tested whether hRpn13 is required in cells for RA190-induced accumulation of ubiquitinated proteins at the proteasome, and indeed find that it is (new Figure 5f).

In terms of the second comment related to INO80, Uch37 is inactivated by this binding partner. The function of Uch37 in this context is not known; thus, there is no strategy currently available to dissect the impact of RA190 in the INO80 context.

2. Figure 3G. Does amount of proteasome-bound UCH37 change upon Rpt3 IP in the presence of RA190? Is it consistent with data in Figures 5A and 5B? How about levels of 20S subunits? What is the functional consequence of RA190 binding to DEUBAD domain?

Response: To address this comment directly, we performed a glycerol gradient fractionation experiment in HCT116 cells treated with RA190 or DMSO (as a control). No change was observed for the presence of Uch37 or 20S subunit $\beta 5$ in proteasome fractions following RA190 treatment; this data is now included in the manuscript at Figure 5c. We also performed the suggested IP experiment and did not observe a change in the amount of detected Uch37 or $\beta 5$ immunoprecipitated with Rpt3 following RA190 treatment; see below.

The protein levels of Uch37 at the proteasome was not changed by RA190. HCT116 cells were treated with 1 μ M RA190 or DMSO for 24 hours and then were cross-linked with DSP or DMSO as a control. Cell lysates were immunoprecipitated with anti-hRpt3 antibodies and immunoprobed with anti-Uch37 or anti- $\beta 5$ antibodies.

These results are consistent with Figure 5a and b.

We think that the reaction of RA190 with hRpn13 C357 reflects its promiscuity rather than a site of functional consequence. We have addressed this comment by adding the following to the Discussion section.

We report that RA190 reacts with hRpn13 DEUBAD domain C357 (Fig. 3c,e); however, this cysteine is directed away from the Uch37-binding surface and does not appear to effect Uch37 interaction with hRpn13 (Fig. 5a,b) or the proteasome (Fig. 5c). Therefore, we do not expect this cysteine to be important for RA190-induced apoptosis, but rather propose that RA190 reaction at this site is a reflection of promiscuity towards exposed cysteine residues.

3. Kd for hRpn2:hRpn13 Pru binding differs when hRpn2 (940-953) or hRpn2 (916-953) are used. Since binding is stronger with the larger fragment, it seems that additional residues contribute to the binding in the larger hRpn2 fragment. It would be beneficial to explain/model/determine it.

Response: In this manuscript, we report a K_d of 27 ± 10 nM for the binding of hRpn2 (940-953) to hRpn13 Pru and published a value of 12 ± 3 nM. Since this difference is almost within error, we expect it to originate from technical issues, rather than additional interactions between amino acids 916-939 of hRpn2. It is worth noting that the K_d value reported in this manuscript for hRpn13 binding to hRpn2 (940-953) as measured by fluorescence polarization is 14.7 ± 0.6 nM.

4. I have a conceptual problem dealing with the fact that 2 previous publications by the same group have basically shown the contradictory data than data presented in this manuscript. I think that the authors

should provide more elaborate explanation why the previous results were obtained, further strengthening the correctness of the current finding.

Response: We believe that the Reviewer is referring to the *Cancer Cell* manuscript that we published in 2013 with Dr. Richard Roden demonstrating that hRpn13 C88 is targeted by RA190 and also a 2016 publication from Dr. Ken Anderson's laboratory, which indicated that RA190 sensitivity is restored in HCT116 cells deleted of hRpn13 upon over-expression of hRpn13 wild-type but not C88A protein.

In our 2013 publication, we revealed that the RA190 class of molecules restrict cancer growth in mice xenograft models of multiple myeloma and ovarian cancer and that they do so through a mechanism involving inhibition of the ubiquitin-proteasome pathway. We used NMR to identify an RA190 interaction surface on hRpn13 and MS to find that C88 of hRpn13 is targeted covalently. These experiments were performed following sample dialysis to remove unbound RA190. Under these conditions, interaction with the hRpn13 Pru domain and not DEUBAD domain was preserved. We now appreciate that reaction with RA190 is reversible and that interaction with the Pru domain in that experiment was preserved most likely because of the surface proximal to C88, which provides favorable contacts to RA190 but also binds to hRpn2/proteasomes. Everything in this earlier manuscript is reproducible when the experiments are conducted in the identical manner. We agree that this point may not have been clear in the original submission and now include the following explanations in our text.

*We previously demonstrated that RA190 adducts to hRpn13 at the proteasome³⁸, where the Pru domain is apparently inaccessible (Fig. 3a), but were also unable to detect RA190-conjugated DEUBAD domain by NMR in samples that were buffer exchanged by dialysis to remove excess RA190³⁸. To resolve this inconsistency, we used LC-MS to test whether the covalent bond between RA190 and hRpn13 Pru is labile in the presence of hRpn2 (940-953). We incubated 2 μ M hRpn13 Pru with 20 μ M RA190 for one hour at 4°C and acquired an LC-MS spectrum to find unmodified and RA190-adducted hRpn13 Pru domain (**Supplementary Fig. 4b, left spectrum**). We then in parallel added either 10-fold molar excess hRpn2 (940-953) or an equivalent volume of buffer to yield final concentrations of 2 μ M hRpn13 Pru, 20 μ M RA190, with or without 20 μ M hRpn2 (940-953). LC-MS spectra were recorded on these two mixtures after one or 19 hours of incubation at 4°C. A time-dependent reduction in RA190-conjugated hRpn13 Pru was observed by hRpn2 addition, whereas this species increased when only buffer was added (**Supplementary Fig. 4b**). This result indicates that RA190 reacts reversibly with this domain and is displaced by hRpn2. Such reversibility is also reported for b-AP15^{52,53}, which is chemically similar to RA190.*

*We hypothesized that RA190 could be even more labile towards the hRpn13 DEUBAD domain, as the Pru domain provides a binding pocket for RA190 when it is conjugated to C88³⁸. By using optimized conditions, including more stringent removal of reducing agent, more diluted samples, and retaining RA190 in the reaction mixture, we detected RA190 conjugated to hRpn13 DEUBAD (**Fig. 3c,e and Supplementary Fig. 4a**).*

5. There might be alternative explanations why hRpn2-bound hRpn13 does not bind RA190, which cannot be excluded, including the potential extra-proteasomal/non-Rpn2-bound fraction of Rpn13 binding

to RA190. I suggest to include NMR or similar type of data to further study the effect of increasing amounts of RA190 on hRpn2:hRpn13 interaction, in order to further strengthen Figure 3C-D data.

Response: This comment relates to that made above by Reviewer 1. We agree with this comment that RA190 may target extra-proteasomal/non-hRpn2-bound fraction of hRpn13 and have included a new figure with data demonstrating that there is a pool of proteasome-free hRpn13 (**Fig. 5c**).

We tested whether increasing amounts of RA190 affects hRpn13 interaction with hRpn2 by FP, but no inhibition was observed for hRpn13 Pru binding to hRpn2 (940-953) even in the presence of 8000-fold molar excess RA190; this new data is included as **Fig. 3g** (left panel). Moreover, no effect was observed on binding affinity between hRpn13 and hRpn2 (940-953) in the presence of 100 μ M RA190 (**Fig. 3g**, right panel). It is possible that RA190-alkylated newly synthesized hRpn13 is prevented from association with the proteasome, but we were unable to find evidence for this model in *vitro*. It is also possible that hRpn13 performs functions outside of the proteasome that use the RA190-binding site with lower affinity compared to hRpn2. Future studies are needed to address the functional significance of hRpn13 outside of the proteasome. We now include such a discussion in the text.

Minor comments

English should be corrected (see few examples below)

28: a narrow canyon of the ubiquitin-binding hRpn13 Pru domain. Interesting (!), mass

31: hRpn13 Uch37-binding DEUBAD domain and Uch37, rendering it catalytically inactivate (!).

Response: Thank you for catching these grammatical mistakes; much appreciated!

REVIEWERS' COMMENTS:

Reviewer #1 (Remarks to the Author):

The authors have addressed all of the comments made by the reviewers satisfactorily. I believe that this manuscript is suitable for publication. Some of the new data are extremely helpful in clearing up points of ambiguity in the original version of the manuscript. In particular, the demonstration that alkylation with RA-190 does not block Rpn13 from loading into the proteasome is particularly helpful in ruling out the model that RA-190 works through effects on newly synthesized Rpn13. This paper will be considered an important piece of work in this growing area.

Reviewer #2 (Remarks to the Author):

My major concern with the first submission of this paper was that RA190 may not be selective for the Uch37 DUB. Considering that a number of the reactions for enzymes of the ubiquitin system depend on cysteine chemistry, the biochemical data may not be interpretable as RA190-induced inhibition of Uch37, but off-target effects. This concern about mechanism/off-target effects was echoed by the other reviewers.

The authors addressed these issues by knocking out hRpn13 from HCT116 cells using CRISPR-Cas9 technology. Loss of hRpn13 was commensurate with loss of Uch37. Gradient fractionation was used to isolate proteasome components, which were analyzed by IP with Rpt3, followed by protein immunoblotting with different antibodies. Loss of hRpn13 and Uch37 at the proteasome leads to alleviation of RA190-induced accumulation of polyubiquitin chains. These new data point to a Uch37/Rpn13 specific mechanism within cells and within an intact, functional proteasome.

I have two minor concerns. For the general audience, the authors should clarify in the manuscript that the proteasome RP retains functionality with knockdown of Rpn13/Uch37, though this is implied in the Figure 5 model. Also, it is somewhat surprising that Rpn13/Uch37 leads to significant buildup of polyubiquitin chains at the proteasome. I would expect the other DUBs, which are apparently not inhibited by RA190, could compensate to some extent. I think the authors should comment on this.

Reviewer #3 (Remarks to the Author):

In the resubmitted manuscript the authors have answered all of my raised questions by including additional data sets and by successfully clarifying the raised issues in the Discussion section.

I think that the manuscript is now suitable to be accepted to Nature Communications.

Response to Reviewers

Reviewer 1

The authors have addressed all of the comments made by the reviewers satisfactorily. I believe that this manuscript is suitable for publication. Some of the new data are extremely helpful in clearing up points of ambiguity in the original version of the manuscript. In particular, the demonstration that alkylation with RA-190 does not block Rpn13 from loading into the proteasome is particularly helpful in ruling out the model that RA-190 works through effects on newly synthesized Rpn13. This paper will be considered an important piece of work in this growing area.

Response: Thank you for your support of our work and for your invaluable suggestions along the way.

Reviewer 2

My major concern with the first submission of this paper was that RA190 may not be selective for the Uch37 DUB. Considering that a number of the reactions for enzymes of the ubiquitin system depend on cysteine chemistry, the biochemical data may not be interpretable as RA190-induced inhibition of Uch37, but off-target effects. This concern about mechanism/off-target effects was echoed by the other reviewers.

The authors addressed these issues by knocking out hRpn13 from HCT116 cells using CRISPR-Cas9 technology. Loss of hRpn13 was commensurate with loss of Uch37. Gradient fractionation was used to isolate proteasome components, which were analyzed by IP with Rpt3, followed by protein immunoblotting with different antibodies. Loss of hRpn13 and Uch37 at the proteasome leads to alleviation of RA190-induced accumulation of polyubiquitin chains. These new data point to a Uch37/Rpn13 specific mechanism within cells and within an intact, functional proteasome.

Response: Thank you for your insights and for helping us to make this study and manuscript stronger.

I have two minor concerns. For the general audience, the authors should clarify in the manuscript that the proteasome RP retains functionality with knockdown of Rpn13/Uch37, though this is implied in the Figure 5 model.

Response: Thank you for bringing this point to our attention. We have now added the following for clarification.

The presence of ubiquitinated proteins at the proteasome was unaltered by hRpn13 deletion (Fig. 5f, lane 3 versus lane 1), suggesting that proteasome is functional in Δ hRpn13 HCT116 cells.

Also, it is somewhat surprising that Rpn13/Uch37 leads to significant buildup of polyubiquitin chains at the proteasome. I would expect the other DUBs, which are apparently not inhibited by RA190, could compensate to some extent. I think the authors should comment on this.

Response: This is an astute observation; namely if Rpn1/Rpn10 are sufficient without Rpn13 then why can't they function properly with RA190-inhibited Rpn13-Uch37? We are pursuing future experiments to address this question; however, two likely contributing factors are allosteric interactions within the RP and direct interactions with trapped substrates. In particular, it is possible that substrate loading at Rpn13 causes allosteric effects that impact Rpn10/Rpn11 or Rpn1/Usp14. In addition, substrates trapped at Rpn13 may not be able to engage Rpn11 or Usp14 in a manner that prevents trapping at Rpn10 and Rpn1. We have tried to address these complicated points briefly in the following added Discussion paragraph.

Nonetheless, the presence of ubiquitinated proteins stalled at RA190-inhibited hRpn13-Uch37 seems to interfere with the ability of these functional receptors (Rpn1 and Rpn10) to clear ubiquitinated proteins from the proteasome (Fig. 5f). Future experiments are required to understand this finding mechanistically; however, it is possible that Rpn1 and Rpn10 are impacted by the presence and/or occupancy of Rpn13 through allosteric relationships within the RP and/or direct interaction with Rpn13-bound substrates that, in response to Uch37 inhibition, become less dynamic or improperly oriented for deubiquitination.

Reviewer 3

In the resubmitted manuscript the authors have answered all of my raised questions by including additional data sets and by successfully clarifying the raised issues in the Discussion section.

I think that the manuscript is now suitable to be accepted to Nature Communications.

Response: Thank you for your enthusiasm towards this work and for your much valued suggestions.